# Temporal Generalization Estimation in Evolving Graphs

**Bin Lu**[1]**, Tingyan Ma**[1]**, Xiaoying Gan**[1]**, Xinbing Wang**[1]
**Yunqiang Zhu**[2]**, Chenghu Zhou**[2]**, Shiyu Liang**[3]*

[1]Department of Electronic Engineering, Shanghai Jiao Tong University
[2]Institute of Geographic Sciences and Natural Resources Research, Chinese Academy of Sciences
[3]John Hopcroft Center for Computer Science, Shanghai Jiao Tong University
{robinlu1209, xiaokeaiyan, ganxiaoying, xwang8}@sjtu.edu.cn
{zhuyq, zhouch}@igsnrr.ac.cn, lsy18602808513@sjtu.edu.cn

## Abstract

Graph Neural Networks (GNNs) are widely deployed in vast fields, but they often struggle to maintain accurate representations as graphs evolve. We theoretically establish a lower bound, proving that under mild conditions, representation distortion inevitably occurs over time. To estimate the temporal distortion without human annotation after deployment, one naive approach is to pre-train a recurrent model (e.g., RNN) before deployment and use this model afterwards, but the estimation is far from satisfactory. In this paper, we analyze the representation distortion from an information theory perspective, and attribute it primarily to inaccurate feature extraction during evolution. Consequently, we introduce Smart, a straightforward and effective baseline enhanced by an adaptive feature extractor through self-supervised graph reconstruction. In synthetic random graphs, we further refine the former lower bound to show the inevitable distortion over time and empirically observe that Smart achieves good estimation performance. Moreover, we observe that Smart consistently shows outstanding generalization estimation on four real-world evolving graphs. The ablation studies underscore the necessity of graph reconstruction. For example, on OGB-arXiv dataset, the estimation metric MAPE deteriorates from 2.19% to 8.00% without reconstruction.

## 1 Introduction

The rapid rising of Graph Neural Networks (GNN) leads to widely deployment in various applications, e.g. social network, smart cities, drug discovery (Yang & Han, 2023; Lu et al., 2022; Xu et al., 2023). However, recent studies have uncovered a notable challenge: as the distribution of the graph shifts continuously after deployment, GNNs may suffer from the representation distortion over time, which further leads to continuing performance degradation (Liang et al., 2018; Wu et al., 2022; Lu et al., 2023), as shown in Figure 1. This distribution shift may come from the continuous addition of nodes and edges, changes in network structure or the introduction of new features. This issue becomes particularly salient in applications where the graph evolves rapidly over time.

Consequently, a practical and urgent need is to monitor the representation distortion of GNN. An obvious method is to regularly label and test online data. However, constant human annotation is difficult to withstand the rapidly ever-growing evolution of graph after deployment. Therefore, how to proactively estimate the temporal generalization performance without annotation after deployment is a challenging problem.

To solve this problem, a naive way is to collect the generalization changes through partially-observed labels before deployment, and train a recurrent neural network (RNN) (Mikolov et al., 2010) in a supervised manner. However, existing stud-

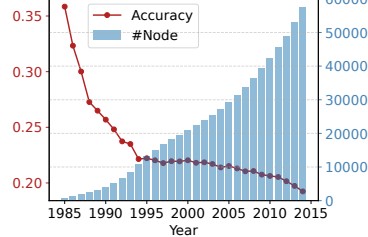

Figure 1: GNN performance continues to decline with the rapid growth over 30 years.

---

*Correspondence to: Shiyu Liang (lsy18602808513@sjtu.edu.cn)

ies (Li et al., 2018; Pareja et al., 2020) have shown that RNN itself have insufficient representation power, thereby usually concatenating a well-designed feature extractor. Unluckily, the representation distortion of static feature extractor still suffers during evolution. Other methods consider measuring the distribution distance between training and testing data through average confidence (Guillory et al., 2021), representation distance Deng & Zheng (2021) and conformal test martingales (Vovk et al., 2021). However, these methods do not directly estimate generalization performance. Meanwhile, graph evolution can be very fast and the evolving graphs are usually very different from their initial states. Hence, to deal with this problem, we propose SMART (Self-supervised teMporAl geneRalization esTimation). Since it is hard to gather label information in a rapid graph evolution after deployment, our SMART resorts to an adaptive feature extractor through self-supervised graph feature and structure reconstruction, eliminating the information gap due to the evolving distribution drift.

We summarize the main contributions as follows:

- We theoretically prove the representation distortion is unavoidable during evolution. We consider a single-layer GCN with Leaky ReLU activation function and establish a lower bound of distortion, which indicates it is strictly increasing under mild conditions. (see Section 2.3)

- We propose a straightforward and effective baseline SMART for this underexplored problem, which is enhanced with self-supervised graph reconstruction to minimize the information loss in estimating the generalization performance in evolving graphs. (see Section 3)

- Under a synthetic Barabási–Albert random graph setting, we refine the lower bound to derive a closed-form generalization error of GNN on whole graph after deployment, and our SMART achieves a satisfied prediction performance. (see Section 4)

- We conduct extensive experiments on four real-world evolving graphs and our SMART consistently shows outstanding performance on different backbones and time spans. The ablation studies indicate the augmented graph reconstruction is of vital importance, e.g., our method reduces the MAPE from 8.00% to 2.19% on OGB-arXiv citation graphs. (see Section 5)

## 2 PRELIMINARY

### 2.1 NOTATIONS

**Graph.** Let $X \in \mathbb{R}^{n \times d}$ denote the feature matrix where each row $x_i \in \mathbb{R}^d$ of the matrix $X$ denotes the feature vector of the node $i$ in the graph. Let $Y \in \mathbb{R}^n$ denote the label matrix where each element $y_i \in \mathbb{R}$ of the matrix $Y$ denotes the label of the node $i$. Let $A \in \{0, 1\}^{n \times n}$ denote the adjacency matrix where the element $a_{ij} = 1$ denotes the existence of an edge between node $i$ and $j$, while $a_{ij} = 0$ denotes the absence of an edge. Let $\tilde{A} = A + I$ denote the self-looped adjacency matrix, while the diagonal matrix $\tilde{D} = I + \text{Diag}(d_1, ..., d_n)$ denotes the self-looped degree matrix, where $d_i = \sum_{j \neq i} A_{ij}$ denotes the degree of node $i$. Let $L = \tilde{D}^{-1}\tilde{A}$ denote a standard normalized self-looped adjacency matrix (Perozzi et al., 2014).

**Graph Evolution Process.** We use $\mathcal{G}_t = (A_t, X_t, Y_t)$ to denote the graph at time $t$. We consider an ever-growing evolution process $\mathcal{G} = (\mathcal{G}_t : t \in \mathbb{N})$, where $\mathbb{N}$ denotes set of all natural numbers. We use $n_t$ to denote the number of nodes in the graph $\mathcal{G}_t$. We use the conditional probability $\mathbb{P}(\mathcal{G}_t | \mathcal{G}_0, ..., \mathcal{G}_{t-1})$ to denote the underlying transition probability of the graph state $\mathcal{G}_t$ at time $t$ conditioned on the history state $\mathcal{G}_0, ..., \mathcal{G}_{t-1}$.

### 2.2 PROBLEM FORMULATION

As shown in Figure 2, before the deployment at time $t_{\text{deploy}}$, we are given with a pre-trained graph neural network $G$ that outputs a label for each node in the graph, based on the graph adjacency matrix $A_k \in \{0, 1\}^{n_k \times n_k}$ and feature matrix $X_k \in \mathbb{R}^{n_k \times d}$ at each time $k$. Additionally, we are given an observation set $\mathcal{D} = \{(A_k, X_k, H_k Y_k)\}_{k=0}^t$, consisting of (1) a series of fully observed graph adjacency matrices $A_0, \cdots, A_t$ and node feature matrices $X_0, \cdots, X_t$; (2) a series of partially observed label vectors $H_0 Y_0, \cdots, H_t Y_t$, where each observation matrix $H_k \in \{0, 1\}^{n_k \times n_k}$ is diagonal. Specifically, the diagonal element on the $i$-th row in matrix $H_k$ is non-zero if and only if the label of the node $i$ is observed.

At each time $\tau$ after deployment at time $t_{\text{deploy}}$, we aim to predict the expected temporal generalization performance of the model $G$ on the state $\mathcal{G}_\tau = (A_\tau, X_\tau, Y_\tau)$, i.e., predicting the generalization error $\mathbb{E}\left[\ell(G(A_\tau, X_\tau), Y_\tau)|A_\tau, X_\tau\right]$, given a full observation on the adjacency matrix $A_\tau$ and feature matrix $X_\tau$. Obviously, the problem

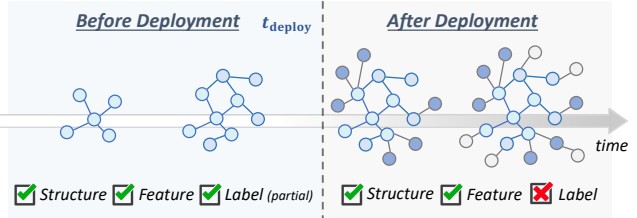

Figure 2: Illustration for generalization estimation.

is not hard if we have a sufficient amount of labeled samples drawn from the distribution $\mathbb{P}\left(Y_\tau|\mathcal{G}_0, ..., \mathcal{G}_{\tau-1}, A_\tau, X_\tau\right)$. However, it is costly to obtain the human annotation on the constant portion of the whole graph after deployment, especially many real-world graphs grow exponentially fast. Therefore, the problem arises if we try to design a post-deployment testing performance predictor $\mathcal{M}$ without further human annotations after deployment. More precisely, we try to ensure that the accumulated performance prediction error is small within a period of length $T - t_{\text{deploy}}$ after deployment time $t_{\text{deploy}}$, where the accumulated performance prediction error is defined as

$$\mathcal{E}(\mathcal{M}; G) \triangleq \mathbb{E}_{\mathcal{A}, \mathcal{X}, \mathcal{Y}} \left[ \sum_{\tau=t_{\text{deploy}}+1}^{T} \left( \mathcal{M}(A_\tau, X_\tau) - \ell(G(A_\tau, X_\tau), Y_\tau) \right)^2 \right], \tag{1}$$

where the random matrix sequence $\mathcal{A} = (A_{t_{\text{deploy}}+1}, ..., A_T)$, sequence $\mathcal{X} = (X_{t_{\text{deploy}}+1}, ..., X_T)$ and sequence $\mathcal{Y} = (Y_{t_{\text{deploy}}+1}, ..., Y_T)$ contain all adjacency matrices, all feature matrices and all label vectors from time $t_{\text{deploy}} + 1$ to $T$ separately.

## 2.3 UNAVOIDABLE REPRESENTATION DISTORTION AS GRAPH EVOLVES

One may ask, then, what impact the pre-trained graph neural network may have on the node representation over time? In this subsection, we show that as the graph evolves, the distortion of representation is unavoidable, which might further lead to potential performance damage.

First, as shown in Figure 3, we plot the performance variation on the OGB-arXiv dataset (Hu et al., 2020a) with 30 different GNN checkpoints. We observe that GNN prediction performance continues to deteriorate over time. Additionally, we prove that the representation distortion strictly increasing with probability one. Before presenting the results, we first present the following several assumptions.

**Assumption 1** (Graph Evolution Process). *The initial graph $\mathcal{G}_0 = (A_0, X_0, Y_0)$ has $n$ nodes. (1) We assume that the feature matrix $X_0$ is drawn from a continuous probability distribution supported on $\mathbb{R}^{n \times d}$. (2) At each time $t$ in the process, a new node indexed by $n + t$ appears in the graph. We assume that this node connects with each node in the existing graph with a positive probability and that edges in the graph do not vanish in the process. (3) We assume that the feature vector $x_{n+t}$ has a zero mean conditioned on all previous graph states, i.e., $\mathbb{E}[x_{n+t}|\mathcal{G}_0, ..., \mathcal{G}_{t-1}] = \mathbf{0}_d$ for all $t \geq 1$.*

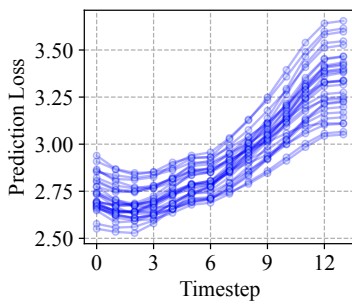

Figure 3: Test loss changes over time of 30 pre-trained GNN checkpoints on OGB-arXiv dataset.

**Remark 1.** Assumption 1 states the following: (1) For any given subspace in the continuous probability distribution, the probability that $X_0$ appears is zero. (2) The ever-growing graph assumption is common in both random graph analysis, like the Barabási-Albert graph (Albert & Barabási, 2002), and real-world scenarios. For example, in a citation network, a paper may have a citation relationships with other papers in the network, and this relationships will not disappear over time. Similarly, the purchasing relationships between users and products in e-commerce trading networks are the same. (3) The zero-mean assumption always holds, as it is a convention in deep learning to normalize the features.

In this subsection, we consider the pre-trained graph model as a single-layer GCN with Leaky ReLU activation function parameterized as $\theta$. The optimal parameters for the model are $\theta^*$. However, due

to the stochastic gradient descent in optimization and quantization of models, we always obtain a sub-optimal parameter drawn from a uniform distribution $U(\theta^*, \xi)$. We define the expected distortion of the model output on node $i$ at time $t$ as the expected difference between the model output at time $t$ and time zero on the node $i$, i.e., $\ell_t(i) = \mathbb{E}_{\theta, \mathcal{G}_t} \left[ |f_t(i; \theta) - f_0(i; \theta)|^2 \right]$.

**Theorem 1.** *If $\theta$ is the vectorization of the parameter set $\{(a_j, W_j, b_j)\}_{j=1}^N$ and its $i$-th coordinate $\theta_i$ is drawn from the uniform distribution $U(\theta_i^*, \xi)$ centering at the $i$-th coordinate of the vector $\theta_i^*$, the expected deviation $\ell_\tau(i)$ of the perturbed GCN model at the time $\tau \geq 0$ on the node $i \in \{1, ..., n\}$ is lower bounded by*

$$\ell_\tau(i) \geq \phi_\tau(i) \triangleq \frac{N\beta^2\xi^4}{9} \mathbb{E}\left[ \left( \frac{1}{d_\tau(i)} - \frac{1}{d_0(i)} \right)^2 \left\| \sum_{k \in \mathcal{N}_0(i)} x_k \right\|_2^2 \right]$$

*where the set $\mathcal{N}_0(i)$ denotes the neighborhood set of the node $i$ at time 0, $\beta$ is is the slope ratio for negative values in Leaky ReLU. Additionally, $\phi_\tau(i)$ is strictly increasing with respect to $\tau \geq 1$.*

**Remark 2.** Proofs of Theorem 1 can be found in Appendix B. This theorem shows that for any node $i$, the expected distortion of the model output is strictly increasing over time, especially when the width $N$ of the model is quite large in the current era of large models. Note that the continuous probability distribution and the ever-growing assumption in Assumption 1 are necessary. Otherwise, $\left\| \sum_{k \in \mathcal{N}_0(i)} x_k \right\|_2^2$ might be zero for some choice of $X_0$, which would lead to the lower bound being zero for all $\tau$ and further indicate that the lower bound is not strictly increasing.

## 3 METHODOLOGY

### 3.1 A NAIVE WAY OF ESTIMATING GENERALIZATION LOSS

Before deployment, we are given an observation set $\mathcal{D} = \{(A_k, X_k, H_k Y_k)\}_{k=0}^{t_{\text{deploy}}}$. Now, we want to construct a temporal generalization loss estimator that takes the adjacency matrices and feature matrices as its input and predicts the difference between the outputs of the graph neural network $G$ and the observed true labels at time $k$, i.e., $\ell_k = \ell(H_k G(A_k, X_k), H_k Y_k)$.

In order to capture the temporal variation, we adopt a recurrent neural network-based model $\mathcal{M}(\cdot; \theta_{RNN})$ to estimate generalization loss in the future. The RNN model sequentially takes the output of the GNN $G(A_k, X_k)$ as its input and outputs the estimation $\hat{\ell}_k$ at time $k$. To enhance the representation power of RNN, we usually add a non-linear feature extractor $\varphi$ to capture the principal features in the inputs and the RNN model becomes

$$\begin{bmatrix} \hat{\ell}_k \\ h_k \end{bmatrix} = \begin{bmatrix} \mathcal{M}_\ell(\varphi \circ G(A_k, X_k), h_{k-1}) \\ \mathcal{M}_h(\varphi \circ G(A_k, X_k), h_{k-1}) \end{bmatrix},$$

where $h_k$ denotes the hidden state in the RNN, transferring the historical information. During training, we are trying to find the optimal model $\mathcal{M}^*, \varphi^*$ such that the following empirical risk minimization problem is solved,

$$(\mathcal{M}^*, \varphi^*) = \arg\min_{\mathcal{M}, \varphi} \sum_{k=0}^{t_{\text{deploy}}} \|\mathcal{M}_\ell(\varphi \circ G(A_k, X_k), h_{k-1}) - \ell(H_k G(A_k, X_k), H_k Y_k)\|^2.$$

After deployment time $t_{\text{deploy}}$, a naive and straightforward way of using the generalization loss estimator is directly applying the model on the graph sequence $(\mathcal{G}_\tau : \tau > t_{\text{deploy}})$. This results in a population loss prediction error $\mathcal{E}_\tau$ at time $\tau$ given by

$$\mathcal{E}_\tau(\mathcal{M}^*, \varphi^*) = \mathbb{E}\|\mathcal{M}_\ell^*(\varphi^* \circ G(A_k, X_k), h_{k-1}) - \ell(G(A_k, X_k), Y_k)\|^2.$$

Before deployment, however, we are not able to get sufficient training frames. This indicates $\varphi^*$ may only have good feature extraction performance on graphs similar to the first several graphs. After deployment, the graphs undergo significant changes, and thereby have unavoidable representation distortion, which makes the generalization estimator perform worse and worse.

## 3.2 INFORMATION LOSS BY REPRESENTATION DISTORTIONS

To further investigate this problem, let us consider the information loss within the input graph series and the output prediction after the GNN is deployed,

$$\text{Information Loss } \triangleq I(\{(A_\tau, X_\tau)\}_{\tau=t_{\text{deploy}}+1}^k, \mathcal{D}; \ell_k) - I(\hat{\ell}_k; \ell_k),$$

where $I(U; V)$ is the mutual information of two random variables $U$ and $V$. The learning process is equivalent to minimizing the above information loss. Furthermore, it can be divided into two parts:

$$\text{Information Loss } = \underbrace{I(\{\varphi \circ G(A_\tau, X_\tau)\}_{\tau=t_{\text{deploy}}+1}^k, \mathcal{D}; \ell_k) - I(\hat{\ell}_k; \ell_k)}_{\text{① Information Loss Induced by RNN}}$$

$$+ \underbrace{I(\{G(A_\tau, X_\tau)\}_{\tau=t_{\text{deploy}}+1}^k, \mathcal{D}; \ell_k) - I(\{\varphi \circ G(A_\tau, X_\tau)\}_{\tau=t_{\text{deploy}}+1}^k, \mathcal{D}; \ell_k)}_{\text{② Information Loss Induced by Representation Distortion}}. \quad (2)$$

**Information Loss Induced by RNN.** The first part is induced by the insufficient representation power of the RNN. Specifically, $\hat{\ell}_k$ is a function of the current state and the hidden state vector $h_{k-1}$. Here, there exists information loss if $h_{k-1}$ cannot fully capture all the historical information. However, this is usually inevitable due to the insufficient representation power of the RNN, limited RNN hidden dimension, and inadequate training data.

**Information Loss Induced by Reprentation Distortion.** The second part indicates the information loss by the representation distortion of $\varphi$. Especially as the graph continues to evolve over time, the information loss correspondingly increases due to the distribution shift. According to the data-processing inequality (Beaudry & Renner, 2012) in information theory, post-processing cannot increase information. Therefore, the second part of Equation 2 holds for any time $\tau$,

$$I(\{G(A_\tau, X_\tau)\}_{\tau=t_{\text{deploy}}+1}^k, \mathcal{D}; \ell_k) - I(\{\varphi \circ G(A_\tau, X_\tau)\}_{\tau=t_{\text{deploy}}+1}^k, \mathcal{D}; \ell_k) \geq 0.$$

The equality holds if and only if $\varphi \circ G(A_\tau, X_\tau)$ is a one-to-one mapping with $G(A_\tau, X_\tau)$. To minimize the information loss and achieve equality, we have to choose a bijective mapping $\varphi$, which further indicates $\min_\phi \sum_\tau \mathbb{E}\|G(A_\tau, X_\tau) - \phi \circ \varphi \circ G(A_\tau, X_\tau)\|_2^2 = 0$. Therefore, to minimize the information loss, it is equivalent to solve the graph representation reconstruction problem. In the next subsection, we propose a reconstruction algorithm to update feature extractor and adapt to the significant graph distribution shift after deployment.

## 3.3 SMART: SELF-SUPERVISED TEMPORAL GENERALIZATION ESTIMATION

In this section, we present our method, SMART, for generalization estimation in evolving graph as shown in Figure 4. After deployment, since there are no human-annotated label, we design the *augmented graph reconstruction* to obtain self-supervised signals to finetune the adaptive feature extractor $\varphi$ at post-deployment time, thereby reducing the information loss during the dynamic graph evolution. Compared with directly graph reconstruction, our method generates more views through data augmentation, which can learn more principal and robust representations.

**Post-deployment Graph Reconstruction.** Given the pre-trained graph model $G$ and evolving graph $(A_k, X_k)$ at time $k$, we first obtain the feature embedding matrix $O_k = G(A_k, X_k)$. In order to capture the evolution property of graphs, we define two reconstruction loss on augmented feature graph, which is denoted as $(\mathcal{T}(A_k), O_k)$, where $\mathcal{T}(\cdot)$ is the structure transformation function, i.e. randomly add or drop edges (You et al., 2020; Rong et al., 2020; Liu et al., 2023).

- **Structure Reconstruction Loss** $\mathcal{L}_s$. First of all, we define the structure reconstruction loss $\mathcal{L}_s$. Given the augmented feature graph $(\mathcal{T}(A_k), O_k)$, we perform reconstruction on adjacency matrix $A_k$. Specifically, it computes the reconstructed adjacency matrix $\hat{A}_k$ by $\hat{A}_k = \sigma(F_k F_k^T), F_k = \varphi(\mathcal{T}(A_k), O_k)$, where $F_k \in \mathbb{R}^{N \times B}$, $\sigma$ is the sigmoid activation function. Here we utilize one-layer graph attention network (GAT) as model $\varphi$ (Velickovic et al., 2018), and it is optimized by binary cross entropy loss between $A_k$ and $\hat{A}_k$ as a link prediction task, i.e. $\mathcal{L}_s = \mathcal{L}_{\text{BCE}}(A_k, \hat{A}_k)$.

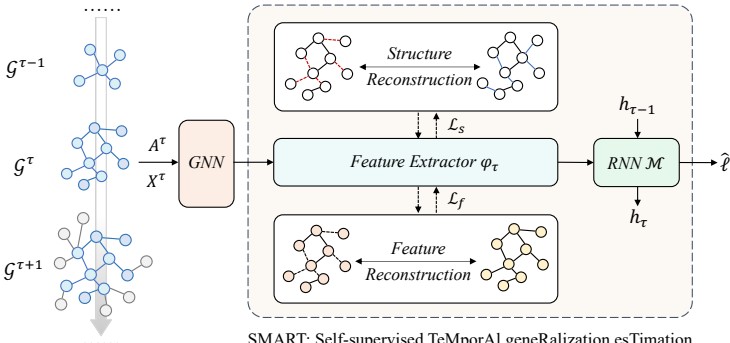

Figure 4: Overview of our proposed SMART for generalization estimation in evolving graph.

- **Feature Reconstruction Loss $\mathcal{L}_f$.** Moreover, we perform the node-level feature reconstruction on the corrupted adjacency matrix $\mathcal{T}(A_k)$. We utilize a single-layer decoder $a$ to obtain the reconstructed feature $\hat{O}_k$ by $\hat{O}_k = F_k a^T, F_k = \varphi(\mathcal{T}(A_k), O_k)$, where $a \in \mathbb{R}^{B \times B}$, and it is optimized by mean squared error loss $\mathcal{L}_f = \|O_k - \hat{O}_k\|^2$.

To sum up, the reconstruction loss $\mathcal{L}_g$ is the composition of structure reconstruction loss $\mathcal{L}_s$ and feature reconstruction loss $\mathcal{L}_f$ as

$$\mathcal{L}_g(\varphi) = \lambda \mathcal{L}_s + (1-\lambda)\mathcal{L}_f, \tag{3}$$

where $\lambda$ is the proportional weight ratio to balance two loss functions.

**Pre-deployment Graph Reconstruction.** We note here that, before deployment, we combine the same graph reconstruction with the supervised learning to improve the performance of feature extraction. Algorithm 1 in Appendix A outlines the pre-deployment warm-up training and post-deployment finetuning process of SMART in detail.

## 4   A CLOSER LOOK AT BARABÁSI–ALBERT RANDOM GRAPH

To theoretically verify the effectiveness of SMART, we first consider the synthetic random graphs, i.e., Barabási–Albert (BA) graph model, $\mathcal{G} = (\mathcal{G}_t : t \in \mathbb{N})$ (Barabási, 2009). Please refer to Appendix E for the details.

**Assumption 2** (Preferential Attachment Evolving Graphs). *Here we consider a node regression task. The initial graph $\mathcal{G}_0$ has $\mathcal{N}_0$ nodes. (1) We assume the node feature matrix $X_k$ is a Gaussian random variable with $\mathcal{N}(0, \mathbf{I}_B)$, where $\mathbf{I}_B \in \mathbb{R}^B$. (2) The node label of each node $i$ is generated by $y_i = d_i^\alpha X_{im}, \alpha \geq 0$, which is satisfied like node degree, closeness centrality coefficients, etc. (3) A single-layer GCN $f(\mathcal{G}) = LXW$ is given as the pre-trained graph model $G$.*

**Theorem 2.** *If at each time-slot $t$, the Barabási–Albert random graph is grown by attaching one new node with $m$ edges that are preferentially attached to existing nodes with high degree. To quantify the performance of GNN, the mean graph-level generalization relative error under the stationary degree distribution $Q$ is determined by*

$$\mathcal{E}_\mathcal{G} = 2m \cdot \frac{t}{\mathcal{N}_0 + t} \cdot (C^2 \mathbb{E}_Q d^{-2\alpha-4} - 2C \mathbb{E}_Q d^{-\alpha-4} + \mathbb{E}_Q d^{-3}),$$

*where $d$ is the degree of nodes. $C = (\mathbb{E}_Q d^{\alpha-1})/(\beta \mathbb{E}_Q d^{-1})$ and $d_i$ is the degree of node $v_i$.*

**Remark 3.** Proofs of Theorem 2 can be found in the Appendix C. This theorem shows that when the node scale $\mathcal{N}_0$ of initial graph $\mathcal{G}_0$ is quite large, the graph-level error loss is approximately linearly related to time $t$ and continues to deteriorate.

To verify the above propositions, we generate a Barabási–Albert (BA) scale-free model with the following setting: the initial scale of the graph is $\mathcal{N}_0 = 1000$, and the evolution period is 180 timesteps. At each timestep, one vertex is added with $m = 5$ edges. The label of each node is the closeness centrality coefficients. The historical training time period is only 9. As we derived in Theorem 2, the actual graph-level generalization error approximates a linear growth pattern. Therefore, we consider to compare SMART with linear regression model to estimate the generalization performance.

**Does Linear Regression Model Work?** We conduct experiments with 10 random seeds and present the empirical results in Figure 5a. (1) Generalization error exhibits linear growth, consistent with our

Table 1: Performance comparison on different Barabási–Albert graph setting. We use Mean Absolute Percentage Error (MAPE) ± Standard Error to evaluate the estimation on different scenarios.

| | Barabási–Albert ($\mathcal{N}_0 = 1000$) | | | Dual Barabási–Albert ($\mathcal{N}_0 = 1000, m_1 = 1$) | | |
|---|---|---|---|---|---|---|
| | $m = 2$ | $m = 5$ | $m = 10$ | $m_2 = 2$ | $m_2 = 5$ | $m_2 = 10$ |
| Linear | 79.2431 | 74.1083 | 82.1677 | 61.8048 | 67.6442 | 38.4884 |
| SMART | $\mathbf{7.1817}_{\pm 1.2350}$ | $\mathbf{4.2602}_{\pm 0.5316}$ | $\mathbf{9.1173}_{\pm 0.1331}$ | $\mathbf{7.9038}_{\pm 1.8008}$ | $\mathbf{3.8288}_{\pm 0.1706}$ | $\mathbf{1.9947}_{\pm 0.1682}$ |

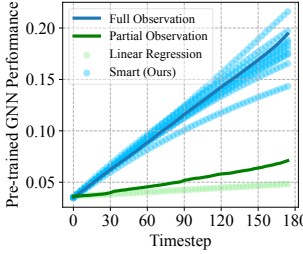 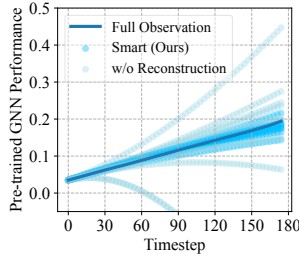 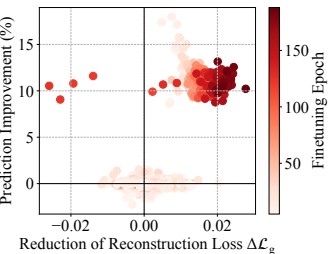

(a) Performance comparison with Linear Regression and SMART.

(b) Ablation study of removing graph reconstruction in SMART.

(c) Prediction improvement during the graph reconstruction.

Figure 5: Experimental results of SMART and its variation on BA random graph.

theorem (the blue solid line). (2) Our SMART method performs significantly well in a long testing period, with a mean prediction percentage error of 4.2602% and collapses into a roughly linear model. (3) However, the linear regression model, based on the first 9-step partial observations (the green solid line), exhibits extremely poor performance. Due to limited human annotation, partial observations cannot accurately represent the performance degradation of the entire graph. We also adjust parameters in the BA graph model and introduced dual BA graph model (Moshiri, 2018) for further experiments (see Table 1). Our proposed SMART model effectively captures the evolutionary characteristics across various settings and significantly outperforms the linear regression model.

**Effectiveness of Graph Reconstruction.** To further validate the effectiveness of graph reconstruction in SMART, we conduct following two experiments. (1) As shown in Figure 5b, we remove the graph reconstruction module and repeat the experiment with 10 random seeds. Due to the temporal distribution shift caused by the graph evolution, the generalization estimation shows significant deviations and instability. (2) We track the intermediate results during post-deployment fine-tuning, i.e. the reduction of reconstruction loss and prediction improvements. As depicted in Figure 5c, in the early stage of reconstruction (scatter points in light color), the prediction performance optimization is fluctuating. As the optimization continues (scatter points in dark color), the prediction performance is effectively boosted and concentrated in the upper-right corner, with an average performance improvement of 10%.

## 5 EXPERIMENTS ON REAL-WORLD EVOLVING GRAPHS

### 5.1 EXPERIMENT SETTINGS

**Datasets.** We use two citation datasets, a co-authorship network dataset and a series of social network datasets for evaluation. The statstics and more details are provided in Appendix F. **OGB-arXiv** (Hu et al., 2020a): A citation network between arXiv papers of 40 subject areas, and we divide the temporal evolution into 14 timesteps. **DBLP** (Galke et al., 2021): A citation network focused on computer science. We divide the temporal evolution into 17 timesteps, and the task is to predict 6 venues. **Pharmabio** (Galke et al., 2021): A co-authorship graph, which records paper that share common authors. We divide the graph into 30 timesteps, and predict 7 different journal categories. **Facebook 100** (Lim et al., 2021): A social network from Facebook of 5 different university: Penn, Amherst, Reed, Johns Hopkins and Cornell. The task is to predict the reported gender.

**Evaluation Metrics.** To evaluate the performance, we adopt following metrics. (1) Mean Absolute Percentage Error (**MAPE**) quantifies the average relative difference between the predicted

generalization loss and the actual observed generalization loss, which is calculated as MAPE $= \frac{1}{T-t_{\text{deploy}}} \sum_{\tau=t_{\text{deploy}}+1}^{T} \left| \frac{\hat{\ell}_\tau - \ell_\tau}{\ell_\tau} \right| \times 100\%$. (2) Standard Error (**SE**) measures the variability of sample mean and estimates how much the sample mean is likely to deviate from the true population mean, which is calculated as $SE = \sigma/\sqrt{n}$, where $\sigma$ is the standard deviation, $n$ is the number of samples. In addition, we conduct the performance comparison on Root Mean Squared Error (RMSE) and Mean Absolute Error (MAE), please refer to Appendix G.1.

**Experiment Configurations.** We adopt the vanilla graph convolution network as the pre-trained graph model $G$, which is trained on the initial timestep of the graph with 10% labeling. During training, we only select the next 3 historical timesteps, where we randomly label 10% of the newly-arrived nodes. The remaining timesteps are reserved for testing, where we have no observations of the label. We run SMART and baselines 20 times with different random seeds.

## 5.2 EXPERIMENTAL RESULTS

**Comparison with Baselines.** In Table 2, we report the performance comparison with two existing baselines, linear regression and DoC (Guillory et al., 2021). Moreover, we compare SMART with a supervised baseline, which requires new node labels and retrain the model on the new data after deployment. We observed that SMART consistently outperforms the other two self-supervised methods (linear regression and DoC) on different evaluation metrics, demonstrating the superior temporal generalization estimation of our methods. On OGB-arXiv dataset, our SMART achieves comparable performance with Supervised.

Table 2: Performance comparison on three academic network datasets and four GNN backbones. We use MAPE ± Standard Error to evaluate the estimation on different scenarios. The complete experimental results can be found in Appendix G.3.

| Dataset | OGB-arXiv ($\downarrow$) | | | | DBLP ($\downarrow$) | | Pharmabio ($\downarrow$) | |
|---|---|---|---|---|---|---|---|---|
| Backbone | Linear | DoC | Supervised | SMART | Linear | SMART | Linear | SMART |
| GCN | 10.5224 | $9.5277_{\pm1.4857}$ | $2.1354_{\pm0.4501}$ | $2.1897_{\pm0.2211}$ | 16.4991 | $3.4992_{\pm0.1502}$ | 32.3653 | $1.3405_{\pm0.2674}$ |
| GAT | 12.3652 | $12.2138_{\pm1.222}$ | $1.9027_{\pm1.1513}$ | $3.1481_{\pm0.4079}$ | 17.6388 | $6.6459_{\pm1.3401}$ | 29.0404 | $1.2197_{\pm0.2241}$ |
| GraphSage | 19.5480 | $20.5891_{\pm0.4553}$ | $1.6918_{\pm0.4556}$ | $5.2733_{\pm2.2635}$ | 23.7363 | $9.9651_{\pm1.4699}$ | 31.7033 | $3.1448_{\pm0.6875}$ |
| TransConv | 14.9552 | $10.0999_{\pm0.1972}$ | $2.0473_{\pm0.5588}$ | $3.5275_{\pm1.1462}$ | 18.2188 | $6.4212_{\pm1.9358}$ | 31.8249 | $2.7357_{\pm1.1357}$ |

**Estimation on Different Test Time Period.** In Table 3, we demonstrate the performance of SMART over time during the evolution of graphs in five social network datasets from Facebook 100. As the evolving pattern gradually deviates from the pre-trained model on the initial graph, generalization estimation becomes more challenging. Consequently, the error in linear estimation increases. However, our SMART method maintains overall stable prediction performance.

Table 3: Performance comparison on five social network datasets in Facebook 100. We divide the test time $T_{\text{test}}$ into 3 periods and evaluate the estimation performance separately.

| Facebook 100 | $[0, T_{\text{test}}/3]$ | | $(T_{\text{test}}/3, 2T_{\text{test}}/3]$ | | $(2T_{\text{test}}/3, T_{\text{test}}]$ | |
|---|---|---|---|---|---|---|
| | Linear | SMART | Linear | SMART | Linear | SMART |
| Penn | 1.9428 | $0.0193_{\pm0.0041}$ | 2.0432 | $0.6127_{\pm0.0307}$ | 2.7219 | $2.2745_{\pm0.0553}$ |
| Amherst | 31.1095 | $1.4489_{\pm0.2450}$ | 49.2363 | $2.8280_{\pm0.9527}$ | 73.5709 | $4.3320_{\pm1.8799}$ |
| Reed | 55.6071 | $0.0453_{\pm0.0020}$ | 65.7536 | $0.0987_{\pm0.0078}$ | 73.6452 | $0.0318_{\pm0.0085}$ |
| Johns Hopkins | 8.1043 | $0.5893_{\pm0.0491}$ | 10.2035 | $0.8607_{\pm0.1661}$ | 11.5206 | $0.9061_{\pm0.2795}$ |
| Cornell | 4.5655 | $0.4663_{\pm0.0275}$ | 8.6622 | $1.0467_{\pm0.0817}$ | 12.3263 | $1.7311_{\pm0.1175}$ |

## 5.3 ABLATION STUDY

To verify the effectiveness of different modules in SMART, we conducted ablation studies on four datasets with the following variants: **(M1) w/o augmented graph reconstruction**: Removing graph reconstruction, using RNN only for estimation. **(M2) w/o RNN**: Replacing RNN with an MLP for processing current step input. **(M3) w/o structure or feature reconstruction**: Removing either structure or feature reconstruction, using the complementary method in post-deployment fine-tuning.

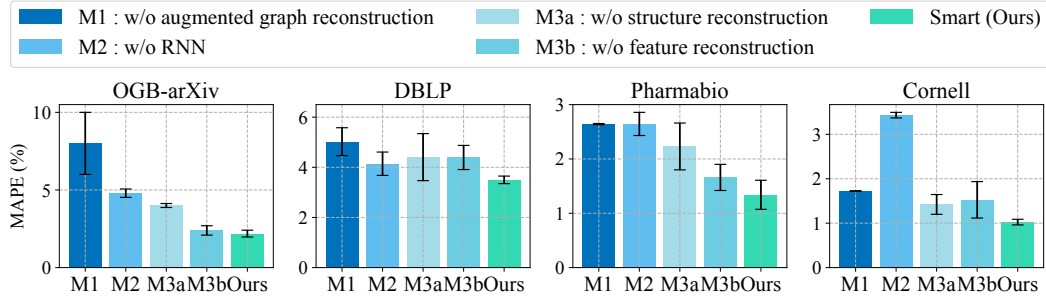

Figure 6: Ablation study on four representative evolving datasets.

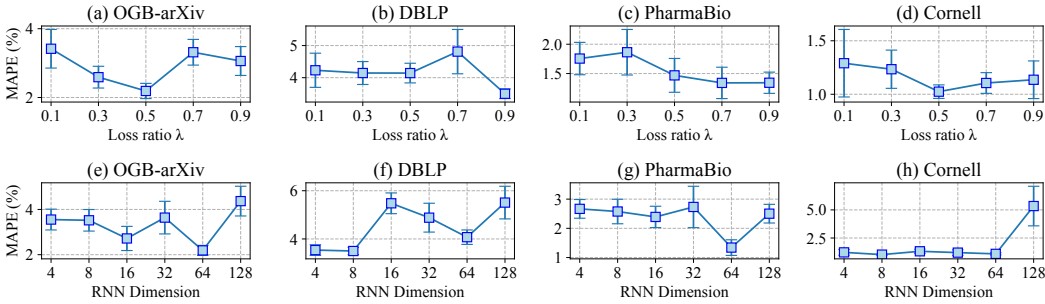

Figure 7: Hyperparamter study on proportional weight ratio $\lambda$ and RNN input dimension.

As seen in Figure 6, we make the following observations: (1) Comparing (M1) with our method, augmented graph reconstruction significantly impacts accurate generalization estimation, particularly evident in the OGB-arXiv and Pharmabio datasets, where the (M1) variant exhibits a substantial performance gap. (2) In the case of (M2), generalization estimation based on historical multi-step information improves prediction accuracy and stability. For instance, in the Cornell dataset, predictions using single-step information result in a larger standard error. (3) As shown by (M3a) and (M3b), removing either of the reconstruction losses leads to a performance decrease in SMART. Since evolving graphs display temporal drift in both structure and features, both graph augmented reconstruction losses are essential for mitigating information loss over time.

### 5.4 HYPERPARAMETER SENSITIVITY

**Proportional Ratio of Two Reconstruction Loss.** We evaluate performance using different weight ratios $\lambda \in \{0, 1, 0.3, 0.5, 0.7, 0.9\}$, as shown in Figure 7(a)-(d). Our method is generally insensitive to the choice of $\lambda$, with $\lambda = 0.5$ being a balanced option in most cases. However, larger $\lambda$ values can yield better results in specific datasets, such as DBLP and PharmaBio, especially when the node features are simple, like one-hot encoding or TF-IDF representations. **Feature Dimension of RNN Input.** We compared RNN feature dimensions ranging from $\{4, 8, 16, 32, 64, 128\}$, as shown in Figure 7(e)-(h). Performance remains stable across four datasets when the feature dimension is set between 4 and 64. However, a significant performance drop occurs on the Cornell dataset when the dimension is set to 128. Setting the RNN feature dimension too high is discouraged for two reasons: (1) As shown in Equation 2, RNN input represents compressed node information over time. To enhance historical information density and effectiveness, the input dimension should be reduced, facilitated by the reconstruction loss. (2) Given limited observation samples during training, reducing the RNN input dimension helps alleviate training pressure.

## 6 CONCLUSIONS

In this paper, we investigate a practical but underexplored problem of temporal generalization estimation in evolving graph. To this end, we theoretically show that the representation distortion is unavoidable and further propose a straightforward and effective baseline SMART. Both synthetic and real-world experiments demonstrate the effectiveness of our methods. Future work involves exploring our methods in more complicated scenarios such as evolving graph with changing label, heterogeneous graphs and spatio-temporal graphs.

ACKNOWLEDGEMENT

This work was partially supported by National Key R&D Program of China (No.2022YFB3904204), NSF China (No. 62306179, 62272301, 42050105, 62020106005, 62061146002, 61960206002), and the Deep-time Digital Earth (DDE) Science Program. Shiyu Liang is also supported by National Natural Science Fund for Excellent Young Scientists Fund Program (Overseas) "Optimizing and Analyzing Deep Residual Networks".

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

## A  ALGORITHM

We illustrate the details of our proposed SMART in Algorithm 1. The learning process of SMART is divided into two stage: pre-deployment warmup training and post-deployment finetuning. Before the deployment, we conduct both the supervised learning on the generalization loss prediction with the partially-observed labels and self-supervised learning on the evolving graph structure. After the deployment, since we no longer have the label information over time, we design two augmented graph reconstruction task as a self-supervised manner to actively finetuning the feature extractor $\varphi$.

---

**Algorithm 1** SMART: Self-supervised Temporal Generalization Estimation

---

**Input:** Pre-trained graph model $G$, observation data $\mathcal{D}$, evolving graph $(A_k, X_k)$ at time k.
**Output:** Update generalization performance predictor $\mathcal{M}$ and $\varphi$ with paramter $\theta = \left[\theta^{\mathcal{M}}, \theta^{\varphi}\right]$.
 1: **while** not converged or maximum epochs not reached **do**    ▷ Pre-deployment Warmup Training
 2:     Compute loss $\mathcal{L}(\theta) = \sum_{\tau=0}^{t_{\text{deploy}}} \|\mathcal{M}_\ell \left(\varphi \circ G(A_\tau, X_\tau), h_{\tau-1}\right) - \ell \left(H_\tau G(A_\tau, X_\tau), H_\tau Y_\tau\right)\|^2$;
 3:     Compute graph self-supervised $\mathcal{L}_g$ via Equation 3;
 4:     Update $\theta \leftarrow \theta + \alpha \nabla_\theta (\mathcal{L} + \mathcal{L}_g)$;
 5: **end while**
 6: **for** $k = t_{\text{deploy}} + 1, \cdots, T$ **do**                    ▷ Post-deployment Finetuning
 7:     Get the newly-arrival graph $(A_k, X_k)$;
 8:     **while** not converged or maximum epochs not reached **do**
 9:         Compute graph self-supervised $\mathcal{L}_g$ via Equation 3;
10:         Update $\theta_k^\varphi \leftarrow \theta_k^\varphi + \beta \nabla_{\theta_k^\varphi} \mathcal{L}_g$;
11:     **end while**
12: **end for**

---

## B  PROOF FOR THEOREM 1

*Proof.* Let $f_\tau(i; \theta)$ denote the output of the GCN on the node $i$ at time $\tau \geq 0$. Therefore, we have

$$f_\tau(i; \theta) = \sum_{j=1}^{N} a_j \sigma \left( \frac{1}{d_\tau(i)} \sum_{k \in \mathcal{N}_\tau(i)} x_k^\top W_j + b_j \right). \tag{4}$$

Thus, the expected loss of the parameter $\theta^*$ on the node $i$ at time $\tau$ is

$$\ell_\tau(i) = \mathbb{E}\left[ \left(f_\tau(i; \theta) - f_0(i; \theta)\right)^2 \right]$$

$$= \mathbb{E}\left[ \left| \sum_{j=1}^{N} a_j \left( \sigma \left( \frac{1}{d_\tau(i)} \sum_{k \in \mathcal{N}_\tau(i)} x_k^\top W_j + b_j \right) - \sigma \left( \frac{1}{d_0(i)} \sum_{k \in \mathcal{N}_0(i)} x_k^\top W_j + b_j \right) \right) \right|^2 \right].$$

Furthermore, recall that each parameter $a_j \sim U(a_j^*, \xi)$ and each element $W_{j,k}$ in the weight vector $W_j$ also satisfies $W_{j,k} \sim U(W_{j,k}^*, \xi)$. Therefore, the differences $a_j - a_j^*$ and $W_{j,k} - W_{j,k}^*$ are all i.i.d. random variables drawn from distribution $U(0, \xi)$. Therefore, we have

$$
\begin{aligned}
\ell_\tau(i) &= \mathbb{E}\left[\left|\left|\sum_{j=1}^{N} a_j \left(\sigma\left(\frac{1}{d_\tau(i)}\sum_{k\in\mathcal{N}_\tau(i)} x_k^\top W_j + b_j\right) - \sigma\left(\frac{1}{d_0(i)}\sum_{k\in\mathcal{N}_0(i)} x_k^\top W_j + b_j\right)\right)\right|\right|^2\right] \\
&= \mathbb{E}\left[\left|\left|\sum_{j=1}^{N} (a_j - a_j^*)\left(\sigma\left(\frac{1}{d_\tau(i)}\sum_{k\in\mathcal{N}_\tau(i)} x_k^\top W_j + b_j\right) - \sigma\left(\frac{1}{d_0(i)}\sum_{k\in\mathcal{N}_0(i)} x_k^\top W_j + b_j\right)\right)\right.\right.\right. \\
&\qquad\left.\left.\left. + \sum_{j=1}^{N} a_j^*\left(\sigma\left(\frac{1}{d_\tau(i)}\sum_{k\in\mathcal{N}_\tau(i)} x_k^\top W_j + b_j\right) - \sigma\left(\frac{1}{d_0(i)}\sum_{k\in\mathcal{N}_0(i)} x_k^\top W_j + b_j\right)\right)\right|\right|^2\right] \\
&= \mathbb{E}\left[\left|\left|\sum_{j=1}^{N} (a_j - a_j^*)\left(\sigma\left(\frac{1}{d_\tau(i)}\sum_{k\in\mathcal{N}_\tau(i)} x_k^\top W_j + b_j\right) - \sigma\left(\frac{1}{d_0(i)}\sum_{k\in\mathcal{N}_0(i)} x_k^\top W_j + b_j\right)\right)\right|\right|^2\right] \\
&\qquad + \mathbb{E}\left[\left|\left|\sum_{j=1}^{N} a_j^*\left(\sigma\left(\frac{1}{d_\tau(i)}\sum_{k\in\mathcal{N}_\tau(i)} x_k^\top W_j + b_j\right) - \sigma\left(\frac{1}{d_0(i)}\sum_{k\in\mathcal{N}_0(i)} x_k^\top W_j + b_j\right)\right)\right|\right|^2\right]
\end{aligned}
$$

The third equality holds by the fact that the differences $(a_j - a_j^*)$'s are all i.i.d. random variables drawn from the uniform distribution $U(0, \xi)$. Therefore, we have

$$
\ell_\tau(i) \geq \mathbb{E}\left[\left|\left|\sum_{j=1}^{N} (a_j - a_j^*)\left(\sigma\left(\frac{1}{d_\tau(i)}\sum_{k\in\mathcal{N}_\tau(i)} x_k^\top W_j + b_j\right) - \sigma\left(\frac{1}{d_0(i)}\sum_{k\in\mathcal{N}_0(i)} x_k^\top W_j + b_j\right)\right)\right|\right|^2\right].
$$

Furthermore, since the differences $(a_j - a_j^*)$ are i.i.d. random variable drawn from the distribution $U(0, \xi)$, we must further have

$$
\begin{aligned}
\ell_\tau(i) &\geq \mathbb{E}\left[\left|\left|\sum_{j=1}^{N} (a_j - a_j^*)\left(\sigma\left(\frac{1}{d_\tau(i)}\sum_{k\in\mathcal{N}_\tau(i)} x_k^\top W_j + b_j\right) - \sigma\left(\frac{1}{d_0(i)}\sum_{k\in\mathcal{N}_0(i)} x_k^\top W_j + b_j\right)\right)\right|\right|^2\right] \\
&= \mathbb{E}\left[\sum_{j=1}^{N} \mathbb{E}[(a_j - a_j^*)^2 | A_\tau, X_\tau]\left|\left|\sigma\left(\frac{1}{d_\tau(i)}\sum_{k\in\mathcal{N}_\tau(i)} x_k^\top W_j + b_j\right) - \sigma\left(\frac{1}{d_0(i)}\sum_{k\in\mathcal{N}_0(i)} x_k^\top W_j + b_j\right)\right|\right|^2\right] \\
&= \frac{\xi^2}{3}\mathbb{E}\left[\sum_{j=1}^{N}\left|\left|\sigma\left(\frac{1}{d_\tau(i)}\sum_{k\in\mathcal{N}_\tau(i)} x_k^\top W_j + b_j\right) - \sigma\left(\frac{1}{d_0(i)}\sum_{k\in\mathcal{N}_0(i)} x_k^\top W_j + b_j\right)\right|\right|^2\right].
\end{aligned}
$$

Furthermore, the leaky ReLU satisfies that $|\sigma(u) - \sigma(v)| \geq \beta|u - v|$. The above inequality further implies

$$
\ell_\tau(i) \geq \frac{\xi^2}{3} \mathbb{E}\left[\sum_{j=1}^N \left|\sigma\left(\frac{1}{d_\tau(i)} \sum_{k \in \mathcal{N}_\tau(i)} x_k^\top W_j + b_j\right) - \sigma\left(\frac{1}{d_0(i)} \sum_{k \in \mathcal{N}_0(i)} x_k^\top W_j + b_j\right)\right|^2\right]
$$

$$
\geq \frac{\beta^2 \xi^2}{3} \mathbb{E}\left[\sum_{j=1}^N \left|\frac{1}{d_\tau(i)} \sum_{k \in \mathcal{N}_\tau(i)} x_k^\top W_j + b_j - \frac{1}{d_0(i)} \sum_{k \in \mathcal{N}_0(i)} x_k^\top W_j - b_j\right|^2\right]
$$

$$
\geq \frac{\beta^2 \xi^2}{3} \mathbb{E}\left[\sum_{j=1}^N \left|\frac{1}{d_\tau(i)} \sum_{k \in \mathcal{N}_\tau(i) \setminus \mathcal{N}_0(i)} x_k^\top W_j + \left(\frac{1}{d_\tau(i)} - \frac{1}{d_0(i)}\right) \sum_{k \in \mathcal{N}_0(i)} x_k^\top W_j\right|^2\right]
$$

$$
\geq \frac{\beta^2 \xi^2}{3} \mathbb{E}\left[\sum_{j=1}^N \left(\left|\frac{1}{d_\tau(i)} \sum_{k \in \mathcal{N}_\tau(i) \setminus \mathcal{N}_0(i)} x_k^\top W_j\right|^2 + \left|\left(\frac{1}{d_\tau(i)} - \frac{1}{d_0(i)}\right) \sum_{k \in \mathcal{N}_0(i)} x_k^\top W_j\right|^2\right)\right]
$$

$$
\geq \frac{\beta^2 \xi^2}{3} \mathbb{E}\left[\sum_{j=1}^N \left|\left(\frac{1}{d_\tau(i)} - \frac{1}{d_0(i)}\right) \sum_{k \in \mathcal{N}_0(i)} x_k^\top W_j\right|^2\right]
$$

Therefore, we have

$$
\ell_\tau(i) \geq \frac{\beta^2 \xi^2}{3} \mathbb{E}\left[\sum_{j=1}^N \left|\left(\frac{1}{d_\tau(i)} - \frac{1}{d_0(i)}\right) \sum_{k \in \mathcal{N}_0(i)} x_k^\top W_j\right|^2\right]
$$

$$
= \frac{\beta^2 \xi^2}{3} \mathbb{E}\left[\sum_{j=1}^N \left|\left(\frac{1}{d_\tau(i)} - \frac{1}{d_0(i)}\right) \sum_{k \in \mathcal{N}_0(i)} x_k^\top (W_j - W_j^* + W_j^*)\right|^2\right]
$$

$$
= \frac{\beta^2 \xi^2}{3} \mathbb{E}\left[\sum_{j=1}^N \left|\left(\frac{1}{d_\tau(i)} - \frac{1}{d_0(i)}\right) \sum_{k \in \mathcal{N}_0(i)} x_k^\top (W_j - W_j^*)\right|^2\right]
$$

$$
+ \frac{\beta^2 \xi^2}{3} \mathbb{E}\left[\sum_{j=1}^N \left|\left(\frac{1}{d_\tau(i)} - \frac{1}{d_0(i)}\right) \sum_{k \in \mathcal{N}_0(i)} x_k^\top W_j^*\right|^2\right],
$$

where the last equality comes from the fact that random vectors $(W_j - W_j^*)$'s are an i.i.d. random variables drawn from the uniform distribution $U(0, \xi)$ and are also independent of the graph evolution process. Therefore, we have

$$
\ell_\tau(i) \geq \frac{N\beta^2 \xi^2}{3} \mathbb{E}\left[\left|\left(\frac{1}{d_\tau(i)} - \frac{1}{d_0(i)}\right) \sum_{k \in \mathcal{N}_0(i)} x_k^\top (W_j - W_j^*)\right|^2\right]
$$

$$
= \frac{N\beta^2 \xi^4}{9} \mathbb{E}\left[\left(\frac{1}{d_\tau(i)} - \frac{1}{d_0(i)}\right)^2 \left\|\sum_{k \in \mathcal{N}_0(i)} x_k\right\|_2^2\right],
$$

where the last equality comes from the fact that each element in the random vector $(W_j - W_j^*)$ is i.i.d. random variable drawn from the uniform distribution $U(0, \xi)$. Since the initial feature matrix $X_0 = (x_1, ..., x_n)$ are drawn from a continuous distribution supported on $\mathbb{R}^d$, we must have with

probability one,

$$\left\|\sum_{k \in \mathcal{N}_0(i)} x_k \right\|_2^2 > 0.$$

Furthermore, we have

$$\mathbb{E}_{\mathcal{G}_\tau}\left[\left(\frac{1}{d_\tau(i)} - \frac{1}{d_0(i)}\right)^2 \bigg| \mathcal{G}_0\right] \geq \left(\mathbb{E}_{\mathcal{G}_\tau}\left[\frac{1}{d_\tau(i)} \bigg| \mathcal{G}_0\right] - \frac{1}{d_0(i)}\right)^2$$

$$= \left(\frac{1}{d_0(i)} - \mathbb{E}_{\mathcal{G}_\tau}\left[\frac{1}{d_\tau(i)} \bigg| \mathcal{G}_0\right]\right)^2$$

To prove $\phi_\tau(i)$ is strictly increasing, it suffices to prove that $\mathbb{E}_{\mathcal{G}_\tau}\left[\frac{1}{d_\tau(i)} \bigg| \mathcal{G}_0\right]$ is decreasing with respect to $\tau$. Since

$$\mathbb{E}_{\mathcal{G}_\tau}\left[\frac{1}{d_\tau(i)} \bigg| \mathcal{G}_0\right] = \mathbb{E}_{\mathcal{G}_\tau}\left[\frac{1}{d_\tau(i)} \bigg| \mathcal{G}_0\right]$$

$$= \int_0^\infty \mathbb{P}\left(\frac{1}{d_\tau(i)} > r \bigg| \mathcal{G}_0\right) dr$$

$$= \int_0^\infty \mathbb{P}\left(d_\tau(i) < \frac{1}{r} \bigg| \mathcal{G}_0\right) dr$$

$$< \int_0^\infty \mathbb{P}\left(d_{\tau-1}(i) < \frac{1}{r} \bigg| \mathcal{G}_0\right) dr,$$

where the last inequality comes from the fact that

$$\mathbb{P}\left(d_\tau(i) < \frac{1}{r} \bigg| \mathcal{G}_0\right) < \mathbb{P}\left(d_{\tau-1}(i) < \frac{1}{r} \bigg| \mathcal{G}_0\right).$$

□

## C    PROOF FOR THEOREM 2

*Proof.* Assuming a regression task with a single-layer GCN, we compute mean squared error between prediction and ground truth as the learning objective as follows

$$\min_W \mathbb{E}_X \|LXW - Y\|_2^2 = \min_W \mathbb{E}_X \left((LXW)^T \cdot LXW - 2(LXW)^T Y + \|Y\|_2^2\right) \tag{5}$$

$$= \min_W \mathbb{E}_X \left(W^T X^T L^T LXW - 2W^T X^T L^T Y + \|Y\|_2^2\right). \tag{6}$$

Since $D^{-2}$ is a diagonal matrix, $(L^T L)_{ij} = (A^T D^{-2} A)_{ij} = a_i^T D^{-2} a_j$, where $a_i$ is the i-th row in matrix $A$. Each node feature is independently Gaussian distributed.

When $i \neq j$,

$$\mathbb{E}_X [X^T L^T LX]_{ij} = 0 \tag{7}$$

When $i = j$,

$$\mathbb{E}_X [X^T L^T LX]_{ii} = \mathbb{E}_X (x_i^T L^T L x_i) = \mathbb{E}_X \left(\sum_{j=1}^n \sum_{m=1}^n x_{ij}(L^T L)_{jm} x_{mi}\right) \tag{8}$$

Similarly, when and only when $j = m$, $\mathbb{E}_X [X^T L^T LX]_{ii} \neq 0$

$$\mathbb{E}_X \left[ X^T L^T L X \right]_{ii} = \sum_{j=1}^{n} \mathbb{E}_X \left( x_{ij}^2 \right) (L^T L)_{jj} = \sum_{j=1}^{n} (L^T L)_{jj} \cdot I \tag{9}$$

where $\sum_{j=1}^{n} (L^T L)_{jj} = \sum_{j=1}^{n} ((D^{-1}A)^T D^{-1}A)_{jj} = \sum_{j=1}^{n} (A^T D^{-2} A)_{jj} = \sum_{j=1}^{n} \frac{1}{d_j} \triangleq \beta$

Consequently, the learning objective is equal to

$$\min_W \mathbb{E}_X \|LXW - Y\|_2^2 = \min_W \beta \cdot W^T W - 2W^T \mathbb{E}_X \left( X^T L^T Y \right) + \|Y\|^2 \tag{10}$$

Meanwhile, the optimal parameter of GCN equals to $W^* = \frac{1}{\beta} \mathbb{E}_X \left( X^T L^T Y \right)$.

$$W_i^* = \frac{1}{\beta} \mathbb{E}_X \left[ X^T L^T Y \right]_i \tag{11}$$

$$= \frac{1}{\beta} \mathbb{E}_X \sum_{m=1}^{N} (X^T L^T)_{im} Y_m \tag{12}$$

$$= \frac{1}{\beta} \mathbb{E}_X \sum_{m=1}^{N} (LX)_{mi} d_m^\alpha X_{mk} \tag{13}$$

$$= \frac{1}{\beta} \mathbb{E}_X \sum_{m=1}^{N} \sum_{s=1}^{N} L_{ms} X_{si} d_m^\alpha X_{mk} \tag{14}$$

$$= \frac{1}{\beta} \mathbb{E}_X \sum_{m=1}^{N} L_{mm} X_{mi} d_m^\alpha X_{mk} \tag{15}$$

Therefore, only if $i = k$, $W_k^* = \frac{1}{\beta} \sum_{m=1}^{N} L_{mm} d_m^\alpha = \frac{1}{\beta} \sum_{m=1}^{N} d_m^{\alpha-1} \triangleq C$, and otherwise $W_i^* = 0$.
For any given node $v_i$, we have

$$\varepsilon_i = \frac{\mathbb{E}\|(LXW^*)_i - Y_i\|_2^2}{\mathbb{E}\|Y_i\|_2^2} = \frac{\mathbb{E}(LXW^*)_i^2 - 2(LXW^*)_i Y_i + \|Y_i\|^2}{\mathbb{E}\|Y_i\|_2^2} \tag{16}$$

To be specific,

$$(LXW^*)_i = \sum_{j=1}^{d} (LX)_{ij} W_j^* = (LX)_{ik} W_k^* = C \sum_{s=1}^{N} L_{is} X_{sk} \tag{17}$$

$$(LXW^*)_i Y_i = C \sum_{s=1}^{N} L_{is} X_{sk} d_i^\alpha X_{ik} = C d_i^\alpha L_{ii} = C d_i^{\alpha-1} \tag{18}$$

$$\mathbb{E}(LXW^*)_i^2 = \mathbb{E}[C \sum_{s=1}^{N} L_{is} X_{sk}]^2 = \mathbb{E}[C^2 \sum_{s=1}^{N} (L_{is} X_{sk})^2] = C^2 \sum_{s=1}^{N} L_{is}^2 = C^2 \frac{1}{d_i} \tag{19}$$

$$\mathbb{E}\|Y_i\|^2 = \mathbb{E} d_i^{2\alpha} X_{ik}^2 = d_i^{2\alpha} \tag{20}$$

Therefore, plugging to Eq. 16

$$\varepsilon(d) = \frac{C^2 d^{-1} - 2C d^{\alpha-1} + d^{2\alpha}}{d^{2\alpha}} \tag{21}$$

$$= C^2 d^{-2\alpha-1} - 2C d^{-\alpha-1} + 1 \tag{22}$$

We denote the error of node $i$ as $\epsilon_i$. Therefore, the overall error of graph $\mathcal{G}$ under the stationary degree distribution $Q$ is calculated as $\mathcal{E}_\mathcal{G} = \mathbb{E}_Q \varepsilon(d)$. Here we assume the inherent graph model

follows the Barabási–Albert model (Albert & Barabási, 2002), where the probability of node degree equals to

$$P(d) = 2m^2 \cdot \frac{t}{\mathcal{N}_0 + t} \cdot \frac{1}{d^3}. \tag{23}$$

$\mathcal{N}_0$ is the initial scale of graphs, $m$ is the newly-arrival number of nodes of each time $t$. Consequently, the error of graph $\mathcal{G}$ can be further deduced as:

$$\mathcal{E}_{\mathcal{G}} = \mathbb{E}_Q \left[ 2m^2 \cdot \frac{t}{\mathcal{N}_0 + t} \frac{1}{d^3} (C^2 \cdot d^{-2\alpha-1} - 2C \cdot d^{-\alpha-1} + 1) \right] \tag{24}$$

$$= 2m^2 \frac{t}{\mathcal{N}_0 + t} (C^2 \mathbb{E}_Q d^{-2\alpha-4} - 2C \mathbb{E}_Q d^{-2\alpha-4} + \mathbb{E}_Q d^{-3}) \tag{25}$$

$\square$

## D  RELATED WORK

### D.1  DISTRIBUTION SHIFT ESTIMATION

Deep learning models are sensitive to the data distribution (Deng et al., 2022). Especially when these models are deployed online, how to accurately estimate generalization error and make decisions in advance is a crucial issue. Deng & Zheng (2021); Deng et al. (2021) estimate the recogniion performance by learning an accuracy regression model with image distribution statistics. Rabanser et al. (2019) conduct an empirical study of dataset shift by two-sample tests on pre-trained classifier. Hulkund et al. (2022) use optimal transport to identify distribution shifts in classification tasks. Vovk et al. (2021) propose conformal test martingales to detect the change-point of retraining the model. However, all these methods are designed for images, while evolving graph shows significant different characteristics due to its ever-growing topology. To our best of knowledge, we are the first to investigate the temporal generalization estimation in graphs.

### D.2  EVOLVING GRAPH REPRESENTATION

Real world graphs show dynamic properties of both feature and topology continuously evolving over time. Existing research can be categorized into two types: (1) Temporal Graph Snapshots. Intuitively, the evolving graph can be splited into a graph time series, thereby taking a recurrent model for temporal encoding (Shi et al., 2021a; Sankar et al., 2020). DHPE (Zhu et al., 2018) gets a simple dynamic model base on SVD, which only preserves the first-order proximity between nodes. DynamicTriad (Zhou et al., 2018) models the triadic closure process to capture dynamics and learns node embeddings at each time step. EvolveGCN (Pareja et al., 2020) proposes to use RNN to evolve the parameter of GCN , while ROLAND (You et al., 2022) recurrently updates hierarchical node embeddings. (2) Temporal Graph Events. Some other studies convert the graph representation by grouping the node and edges with time steps in a given period (Wang et al., 2021; Trivedi et al., 2019). TREND (Wen & Fang, 2022) borrows the idea of Hawkes temporal point process in time series analysis, capturing both individual and collective temporal characteristics of dynamic link formation. $M^2$DNE (Lu et al., 2019) proposes a novel temporal network embedding with micro- and macro-dynamics, where a temporal attention point process is designed to capture structural and temporal properties at a fine-grained level. Lou et al. (2023) propose to evaluate the cohesiveness of temporal graphs in both topology and time dimensions.

## E  EXPERIMENT DETAILS OF BARABÁSI-ALBERT RANDOM GRAPH

Barabási-Albert (BA) random graph (Barabási, 2009; Albert & Barabási, 2002), also known as the preferential attachment model, is a type of random graph used to model complex networks, particularly those that exhibit scale-free properties. BA graphs are often used to model various real-world networks, such as the World Wide Web, social networks, citation networks, etc.

BA random graph has two important concepts: growth and preferential attachment. Growth means that the number of nodes in the network increases over time. Preferential attachment means that the more connected a node is, the more likely it is to receive new links. Start with an initial graph $\mathcal{G}_0$

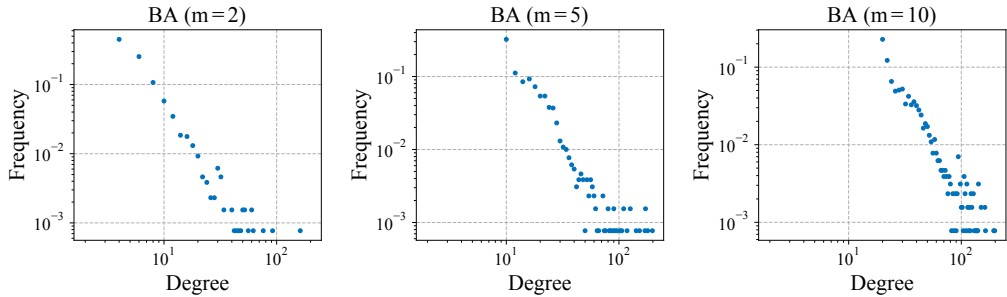

Figure 8: Degree Distribution of BA graphs

that consists of a number of nodes and edges. At each time step $\tau$, a new node is introduced into the graph, and it forms $m$ edges to existing nodes in the graph. The probability that an existing node receives an edge from the new node is proportional to the degree (number of edges) in the current graph $\mathcal{G}_{\tau-1}$. This means that nodes with higher degrees are more likely to receive new edges, exhibiting preferential attachment.

Dual Barabási-Albert (Dual BA) random graph (Moshiri, 2018) is an extension of BA model that better capture properties of real networks of social contact. Dual BA model is parameterized by $\mathcal{N}_0$ inital number of nodes, $1 \leq m_1, m_2, \leq \mathcal{N}_0$ and probability $0 \leq p \leq 1$. For each new vertex, with probability $p$, $m_1$ new edges are added, and with probability $1 - p$, $m_2$ new edges are added. The new edges are added in the same manner as in the BA model.

In our experiment, we adopt above two different BA model as test datasets. We use `barabasi_albert_graph`[1] and `dual_barabasi_albert_graph`[2] implementation provided by NetworkX. The detailed experiment parameter settings (Table 4, Table 5) and the degree distribution (Figure 8, Figure 9) are shown as follows.

Table 4: Parameter setting of BA graph in synthetic experiments.

| Barabási-Albert model | |
| --- | --- |
| $n$ | $m$ |
| 1000 | 2 |
| 1000 | 5 |
| 1000 | 10 |

Table 5: Parameter setting of Dual BA graph in synthetic experiments

| Dual Barabási-Albert model | | |
| --- | --- | --- |
| $n$ | $m_1$ | $m_2$ |
| 1000 | 1 | 2 |
| 1000 | 1 | 5 |
| 1000 | 1 | 10 |

Here, we present the complete experiment results on 4 evaluation metrics (MAPE, RMSE, MAE and Standard Error) in Table 6. Since the DoC method only applies to classification problems using the average confidence, we thus do not compare it with our SMART on the synthetic setting.

## F    REAL-WORLD DATASETS

We use two citation datasets, a co-authorship network dataset and a series of social network datasets to evaluate our model's performance. We utilize inductive learning, wherein nodes and edges that emerge during testing remain unobserved during the training phase. The detailed statistics are shown in Table 7.

---

[1] Please refer to the implementation in `https://networkx.org/documentation/stable/reference/generated/networkx.generators.random_graphs.barabasi_albert_graph.html`

[2] Please refer to the implementation in `https://networkx.org/documentation/stable/reference/generated/networkx.generators.random_graphs.dual_barabasi_albert_graph.html`

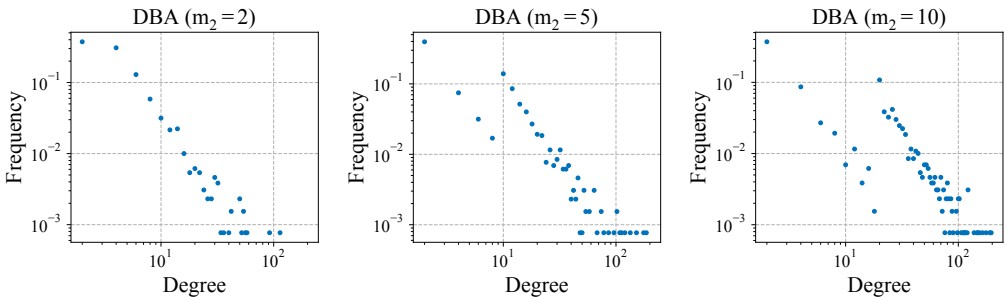

Figure 9: Degree Distribution of DBA graphs

Table 6: Performance comparison on different Barabási–Albert graph setting.

| Barabási-Albert (BA) Random Graph | | Linear | | | SMART | | |
|---|---|---|---|---|---|---|---|
| | | MAPE | RMSE | MAE | MAPE | RMSE | MAE |
| BA ($\mathcal{N}_0 = 1000$) | $m = 2$ | 79.2431 | 0.0792 | 0.0705 | $7.1817_{\pm1.2350}$ | $0.0057_{\pm0.0008}$ | $0.0042_{\pm0.0006}$ |
| | $m = 5$ | 74.1083 | 0.0407 | 0.0398 | $4.2602_{\pm0.5316}$ | $0.0039_{\pm0.0004}$ | $0.0035_{\pm0.0004}$ |
| | $m = 10$ | 82.1677 | 0.1045 | 0.0925 | $9.1173_{\pm0.1331}$ | $0.0077_{\pm0.0010}$ | $0.0071_{\pm0.0009}$ |
| Dual BA ($\mathcal{N}_0 = 1000$, $m_1=1$) | $m_2 = 2$ | 61.8048 | 0.0676 | 0.0615 | $7.9038_{\pm1.8008}$ | $0.0088_{\pm0.0022}$ | $0.0069_{\pm0.0017}$ |
| | $m_2 = 5$ | 67.6442 | 0.0827 | 0.0771 | $3.8288_{\pm0.1706}$ | $0.0049_{\pm0.0013}$ | $0.0040_{\pm0.0010}$ |
| | $m_2 = 10$ | 38.4884 | 0.0298 | 0.0297 | $1.9947_{\pm0.1682}$ | $0.0026_{\pm0.0002}$ | $0.0023_{\pm0.0003}$ |

- **OGB-arXiv** (Hu et al., 2020a): The OGB-arXiv dataset is a citation network where each node represents an arXiv paper, and each edge signifies the citation relationship between these papers. Within this dataset, we conduct node classification tasks, encompassing a total of 40 distinct subject areas. Our experiment spans the years from 2007 to 2020. In its initial state in 2007, OGB-arXiv comprises 4,980 nodes and 6,103 edges. As the graph evolves over time, the citation network boasts 169,343 nodes and 1,166,243 edges. We commence by pretraining graph neural networks on the graph in 2007. Subsequently, we employ data from the years 2008 to 2010 to train our generalization estimation model SMART. Following this training, we predict the generalization performance of the pretrained graph neural network on graphs spanning the years 2011 to 2020.

- **DBLP** (Galke et al., 2021): DBLP is also a citation network and this dataset use the conferences and journals as classes. In our experiment, DBLP starts from 1999 to 2015 with 6 classes. Throughout the evolution of DBLP, the number of nodes increase from 6,968 to 45,407, while the number of edges grow from 25,748 to 267,227. We pretrain the graph neural network on the graph in 1999 and train our model on the next three years. We employ the graph spanning from 2004 to 2015 to assess the performance of our model.

- **Pharmabio** (Galke et al., 2021): Pharmabio is a co-authorship graph dataset, and each node represents a paper with normalized TF-IDF representations of the publication title as its feature. If two papers share common authors, an edge is established between the corresponding nodes. We conduct node classification tasks on this dataset, comprising a total of seven classes, with each class representing a journal category. The range of Pharmabio is 1985 to 2016. The pretrained graph neural network is based on the graph of the year 1985 with 620 nodes and 57,559 edges. Then we train our estimation model by using graph data from 1986 to 1988. We evaluate our model on consecutive 26 years starting form 1989. At the last year 2016, the graph has evolved to 2,820 nodes with 3,005,421 edges.

- **Facebook 100** (Lim et al., 2021): Facebook 100 is a social network which models the friendship of users within five university. We perform binary node classification on this dataset, with the classes representing the gender of the users. Among these datasets, Amherst, Reed and Johns Hopkins are of smaller scale, while Penn and Cornell are larger in size. We sequentially evaluate

Table 7: Overview of real-world evolving graph datasets

| Datasets | OGB-arXiv | DBLP | Pharmabio | Facebook 100 | | | | |
|---|---|---|---|---|---|---|---|---|
| | | | | Penn | Amherst | Reed | Johns Hopkins | Cornell |
| #Node (Start) | 4980 | 6968 | 620 | 5385 | 107 | 85 | 368 | 1360 |
| #Node (End) | 169343 | 45407 | 57559 | 38815 | 2032 | 865 | 4762 | 16822 |
| #Edge (Start) | 6103 | 25748 | 2820 | 47042 | 192 | 188 | 1782 | 8148 |
| #Edge (End) | 1166243 | 267227 | 3005421 | 2498498 | 157466 | 31896 | 338256 | 1370660 |
| Time Span | 14 | 17 | 30 | 18 | 11 | 13 | 14 | 14 |
| #Class | 40 | 6 | 7 | 2 | 2 | 2 | 2 | 2 |

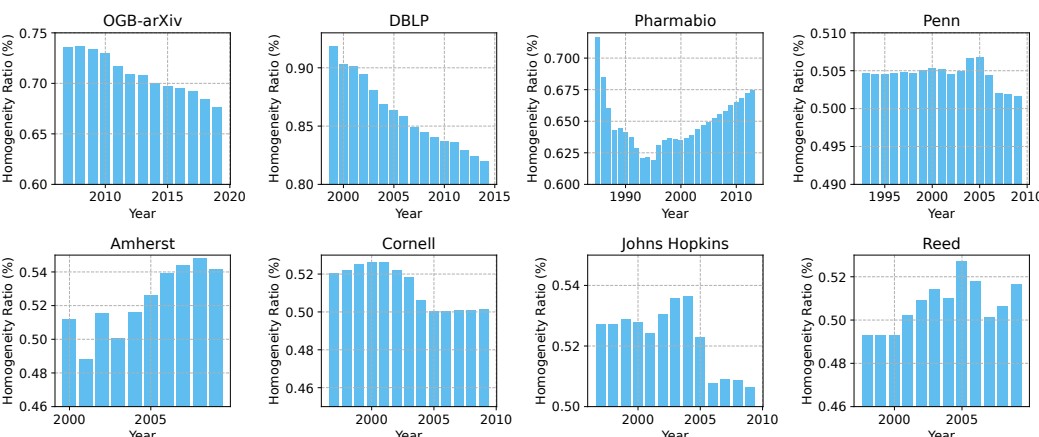

Figure 10: Homophily ratio analysis on different datasets.

our model's adaptability to datasets of different scales. All these datasets end in the year of 2010 with the number of nodes varying from 865 to 38,815 and edges from 31,896 to 2,498,498.

In addition, research on the homogeneity and heterogeneity of graphs has received widespread attention in recent years. In the context of graph data, homogeneity and heterogeneity refer to whether the nodes and edges in the graph are of the same type (homogeneous) or different types (heterogeneous). For example, in a social network, a homogeneous graph might represent individuals (nodes) and friendships (edges), where all nodes and edges are of the same type. On the other hand, a heterogeneous graph could represent individuals, events, and organizations as different types of nodes, with edges representing relationships like "attended," "organized," or "works for." In our work, we do not limit or pre-define the homogeneity and heterogeneity in the data. Moreover, we calculate the homophily ratio of nodes on different experiment datasets, as shown in Figure 10. According to the distribution of homogeneity, there is a significant difference in the proportion of homogeneity among different datasets, and varies greatly over time.

## G ADDITIONAL EXPERIMENTAL RESULTS OF SMART

In this section, we provide a comprehensive illustration of the experiment details for temporal generalization estimation in evolving graphs, including the experiment settings, implementation details and completed results.

### G.1 EXPERIMENT SETTINGS

**Baselines.** We compare the performance of SMART with following three baselines.

- **Linear**: Linear regression is a straightforward statistical learning methods for modeling the relationship between a scalar and variables. We observe that the performance degradation of Barabási-Albert random graph shows approximate linear process. Therefore, to estimate the

GNN performance degradation on real-world datasets, we adopt linear regression as a baseline to predict the performance changes.

- **DoC**: Differences of confidences (DoC) approach (Guillory et al., 2021) proposes to estimate the performance variation on a distribution that slightly differs from training data with average confidence. DoC yields effective estimation of image classification over a variety of shifts and model architecture, outperforming common distributional distances such as Frechet distance or Maximum Mean Discrepancy.

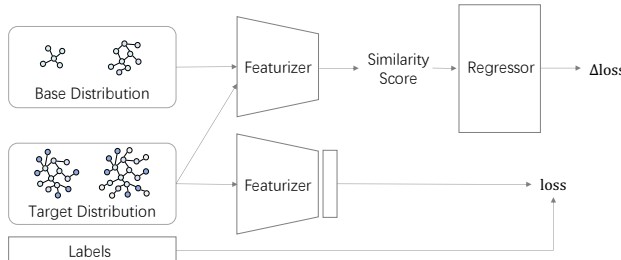

Figure 11: System framework of DoC for temporal generalization estimation in graphs. (Modification of Figure 2 in paper (Guillory et al., 2021))

- **Supervised**: Moreover, we design a comparison method for temporal generalization estimation with supervised signals. Although we are unable to access the labeled data over the time during post-deployment, in our experiment we can acquire new node labels and retrain the model on the new data, which is approximately a upper performance limit of this problem.

**GNN Architectures.** To fully validate and compare the effectiveness of different methods, we select four different graph neural network architectures as follows.

- **GCN** (Kipf & Welling, 2017): GCN is one of the most classic graph neural network architecture, which operates localized first-order approximation of spectral graph convolutions.
- **GAT** (Velickovic et al., 2018) : Graph Attention Network (GAT) leverages attention mechanism to selectively focus on different neighboring nodes of the graph when aggregating information.
- **GraphSage** (Hamilton et al., 2017): GraphSage is a graph neural network architecture designed for scalable and efficient learning on large graphs. It addresses the challenge by sampling and aggregating information from a node's local neighborhood.
- **TransformerConv** (Shi et al., 2021b): In light of the superior performance of Transformer in NLP, TransformerConv adopt transformer architecture into graph learning with taking into account the edge features. The Multi-head attention matrix replaces the original normalized adjacency matrix as transition matrix for message passing.

**Evaluation Metrics.** To evaluate the performance, we adopt following four metrics to measure the effectiveness of temporal generalization estimation over time.

- Mean Absolute Percentage Error (**MAPE**) quantifies the average relative difference between the predicted generalization loss and the actual observed generalization loss, which is calculated as

$$\text{MAPE} = \frac{1}{T - t_{\text{deploy}}} \sum_{\tau = t_{\text{deploy}} + 1}^{T} \left| \frac{\hat{\ell}_\tau - \ell_\tau}{\ell_\tau} \right| \times 100\%.$$

- Root Mean Squared Error (**RMSE**) is calculated by taking the square root of the average of the squared differences between predicted and actual generalization loss, which is calculated as

$$\text{RMSE} = \sqrt{\frac{1}{T - t_{\text{deploy}}} \sum_{\tau = t_{\text{deploy}} + 1}^{T} (\hat{\ell}_\tau - \ell_\tau)^2}.$$

Table 8: Hyperparameter setting in our experiments

| Datasets | OGB-arXiv | DBLP | Pharmabio | Facebook 100 | | | | |
|---|---|---|---|---|---|---|---|---|
| | | | | Penn | Amherst | Reed | Johns Hopkins | Cornell |
| GNN Layer | 3 | 3 | 2 | 2 | 2 | 1 | 2 | 2 |
| GNN dimension | 256 | 256 | 256 | 256 | 256 | 32 | 256 | 256 |
| RNN Layer | 1 | 1 | 1 | 1 | 1 | 1 | 1 | 1 |
| RNN dimension | 64 | 8 | 64 | 64 | 64 | 32 | 64 | 8 |
| loss lambda | 0.5 | 0.9 | 0.7 | 0.3 | 0.9 | 0.7 | 0.1 | 0.5 |
| learning rate | 0.01 | 0.01 | 0.01 | 0.01 | 0.01 | 0.01 | 0.01 | 0.01 |

- Mean Absolute Error (**MAE**) measures the average absolute difference between the predicted generalization loss and the actual generalization loss, which is calculated as

$$\text{MAE} = \frac{1}{T - t_{\text{deploy}}} \sum_{\tau = t_{\text{deploy}} + 1}^{T} \left| \hat{\ell}_\tau - \ell_\tau \right|.$$

- Standard Error (**SE**) measures the variability of sample mean and estimates how much the sample mean is likely to deviate from the true population mean, which is calculated as $SE = \sigma / \sqrt{n}$, where $\sigma$ is the standard deviation, $n$ is the number of samples.

## G.2 IMPLEMENTATION DETAILS

**Data Labeling.**  To simulate real-world human annotation scenarios, we randomly labeled 10% of the samples during the training of the pre-trained graph neural network model. Prior to deployment, at each time step, we additionally labeled 10% of the newly appearing nodes. After deployment, no additional labeling information was available for newly added nodes. For consistency, we use only the first three frames to obtain few labels for all real-world datasets, which is a relatively small sample size. Further enhancing the labeled data can yield additional improvements in temporal generalization estimation.

**Hyperparameter Setting.**  We use Adam optimizer for all the experiments, and the learning rate for all datasets are uniformly set to be 1e-3. In all experiments, the pre-trained graph neural networks are equipped with batch normalization and residual connections, with a dropout rate set to 0.1. Meanwhile, We employed the ReLU activation function. We set hyperparameter for each datasets and specify the details in Table 8.

**Hardware.**  All the evaluated models are implemented on a server with two CPUs (Intel Xeon Platinum 8336C $\times$ 2) and four GPUs (NVIDIA GeForce RTX 4090 $\times$ 8).

## G.3 EXPERIMENTAL RESULTS

**Comparison with different baselines.**  We conduct performance comparison of SMART and other three baselines on three academic network datasets. As shown in Table 9, we observe a strikingly prediction improvement. Our proposed SMART consistently outperforms linear regression and DoC on different evaluation metrics, which demonstrates the superior temporal generalization estimation of our methods. For example, on Pharmabio datasets, due to its long-term temporal evolution, the vanilla linear regression and DoC shows inferior prediction due to the severe GNN representation distortion. Our SMART shows advanced performance due to our self-supervised parameter update over the time.

In addition, when comparing with supervised method, due to its continuous acquisition of annotation information and retraining during the testing phase, it is approximately the upper limit of the estimation performance. On OGB-arXiv dataset, our SMART achieves close performance with Supervised, indicating our strong ability to cope with evolving distribution shift. However, on Pharmabio dataset, accurately estimating its generalization performance remains a challenge due to its long-term evolution (30 timesteps). Thereby, temporal generalization estimation is an important and challenging issue, and still deserves further extensive research.

Table 9: Performance comparison of SMART and baselines on three academic network datasets. MAPE, RMSE, MAE and Standard Error are utilized to evaluate the estimation performance on different datasets. The smaller the value, the better the performance.

| Dataset | Metric | Linear | DoC | SMART (Ours) | Supervised |
|---|---|---|---|---|---|
| OGB-arXiv | MAPE | 10.5224 | $9.5277_{\pm 1.4857}$ | $\mathbf{2.1897_{\pm 0.2211}}$ | $2.1354_{\pm 0.4501}$ |
| | RMSE | 0.4764 | $0.3689_{\pm 0.0400}$ | $\mathbf{0.1129_{\pm 0.0157}}$ | $0.0768_{\pm 0.0155}$ |
| | MAE | 0.4014 | $0.3839_{\pm 0.0404}$ | $\mathbf{0.0383_{\pm 0.0083}}$ | $0.0218_{\pm 0.0199}$ |
| DBLP | MAPE | 16.4991 | $4.3910_{\pm 0.1325}$ | $\mathbf{3.4992_{\pm 0.1502}}$ | $2.5359_{\pm 0.4282}$ |
| | RMSE | 0.5531 | $0.1334_{\pm 0.0058}$ | $\mathbf{0.1165_{\pm 0.0444}}$ | $0.0914_{\pm 0.0175}$ |
| | MAE | 0.431 | $0.1162_{\pm 0.0310}$ | $\mathbf{0.0978_{\pm 0.0344}}$ | $0.0852_{\pm 0.0038}$ |
| Pharmabio | MAPE | 32.3653 | $8.1753_{\pm 0.0745}$ | $\mathbf{1.3405_{\pm 0.2674}}$ | $0.4827_{\pm 0.0798}$ |
| | RMSE | 0.7152 | $0.1521_{\pm 0.0014}$ | $\mathbf{0.0338_{\pm 0.0136}}$ | $0.0101_{\pm 0.0015}$ |
| | MAE | 0.6025 | $0.1521_{\pm 0.0013}$ | $\mathbf{0.0282_{\pm 0.0120}}$ | $0.0088_{\pm 0.0026}$ |

Table 10: Performance comparison of different GNN architectures, including GCN, GAT, Graph-SAGE and TransformerConv. We use MAPE ± Standard Error to evaluate the estimation on different scenarios.

| Dataset | Method | GNN architecture | | | |
|---|---|---|---|---|---|
| | | GCN | GAT | GraphSAGE | TransformerConv |
| OGB-arXiv | Linear | 10.5224 | 12.3652 | 19.5480 | 14.9552 |
| | DoC | $13.5277_{\pm 1.4857}$ | $12.2138_{\pm 1.222}$ | $20.5891_{\pm 0.4553}$ | $10.0999_{\pm 0.1972}$ |
| | SMART | $\mathbf{2.1897_{\pm 0.2211}}$ | $\mathbf{3.1481_{\pm 0.4079}}$ | $\mathbf{5.2733_{\pm 2.2635}}$ | $\mathbf{3.5275_{\pm 1.1462}}$ |
| DBLP | Linear | 16.4991 | 17.6388 | 23.7363 | 18.2188 |
| | DoC | $4.3910_{\pm 0.1325}$ | $13.8735_{\pm 4.1744}$ | $11.9003_{\pm 1.8249}$ | $9.0127_{\pm 2.6619}$ |
| | SMART | $\mathbf{3.4992_{\pm 0.1502}}$ | $\mathbf{6.6459_{\pm 1.3401}}$ | $\mathbf{9.9651_{\pm 1.4699}}$ | $\mathbf{6.4212_{\pm 1.9358}}$ |
| Pharmabio | Linear | 32.3653 | 29.0404 | 31.7033 | 31.8249 |
| | DoC | $8.1753_{\pm 0.0745}$ | $7.4942_{\pm 0.0702}$ | $6.6376_{\pm 0.0194}$ | $5.3498_{\pm 0.2636}$ |
| | SMART | $\mathbf{1.3405_{\pm 0.2674}}$ | $\mathbf{1.2197_{\pm 0.2241}}$ | $\mathbf{3.1448_{\pm 0.6875}}$ | $\mathbf{2.7357_{\pm 1.1357}}$ |

**Comparison with different GNN architectures.** To evaluate the applicability of proposed algorithm, we conduct temporal generalization estimation on different GNN architectures. As shown in Table 10, first of all, our SMART shows consistent best performance among different architectures. Besides, for different graph neural network architectures, they tend to capture the structure and feature information from different aspects. Due to the simple model structure of GCN, our SMART shows advanced prediction performance among other architectures, which is also consistent with the theory of classical statistical machine learning.

**Comparison with different time series model.** To capture the temporal drift of GNN generalization variation, we propose an RNN-based method (i.e. LSTM). Moreover, we additionally replace the LSTM with other two time series model Bi-LSTM and TCN as follows:

- **LSTM**: Long Short-Term Memory (LSTM) is a type of recurrent neural network (RNN) architecture designed to address the vanishing gradient problem in traditional RNNs. The ability of LSTMs to selectively remember or forget information makes them well-suited for tasks involving long-term dependencies, and become a standard architecture in the field of deep learning for sequential data.

- **Bi-LSTM** Graves & Schmidhuber (2005): Bidirectional Long Short-Term Memory (Bi-LSTM) is an extension of the traditional LSTM architecture that enhances its ability to capture information from both past and future context.

- **TCN** (Lea et al., 2017): Temporal Convolutional Network (TCN) use 1D convolutional layers to capture dependencies across different time steps in a sequence. Unlike traditional CNNs, TCNs

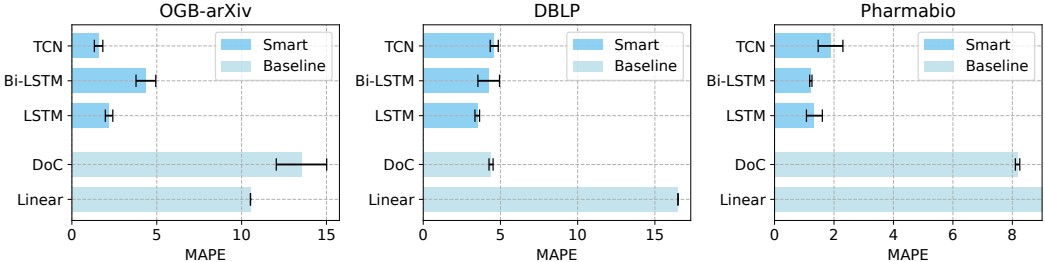

Figure 12: Comparison with different time series model, i.e., LSTM, Bi-LSTM and TCN.

use dilated convolutions to extend the receptive field without significantly increasing the number of parameters.

Different time series methods are worth trying out, since we hope to capture the temporal changes based on a small number of training snapshots. In our experiment, as depicted in Figure 12, our SMART achieves the optimal performance using three different time series models on both OGB-arXiv and Pharmabio datasets. On DBLP dataset, SMART achieve the best performance using LSTM, while SMART equipped with Bi-LSTM and TCN show close performance with DoC. In order to maintain consistency in the experiment, we adopted the same hyperparameter settings. From the perspective of parameter size, the parameter quantity of Bi-LSTM is nearly twice that of LSTM, while the parameter quantity of TCN is very small. It can be seen that as the parameter size increases, the prediction error generally decreases first and then increases. Therefore, it is very important to choose appropriate models and parameter settings based on the complexity of the data.

**Comparison with different graph data augmentation methods.** Self-supervised learning is a popular and effective learning strategy to enrich the supervised signals.

Recent success of self-supervised learning on image datasets heavily rely on various data augmentation methods. Due to the irregular structure of graphs, existing graph augmentation methods can be categorized into topological-based augmentation and attributive-based augmentation. In our work, we adopt three familiar and effective graph data augmentation methods to generate different views for self-supervised graph reconstruction, shown as follows:

- **DropEdge** (Rong et al., 2020): Randomly delete some edges with a certain probability, thereby changing the graph structure as the input of GNN.
- **DropNode** (Papp et al., 2021): Instead of randomly delete some edges, DropNode randomly delete some nodes with a certain probability.
- **Feature Mask** (Hu et al., 2020b): Randomly mask features of a portion of nodes with a certain probability, thereby changing the node feature of graphs.

Figure **??** shows the experimental results, and our SMART shows the best performance on three datasets. Due to our consideration of both structure prediction and feature reconstruction, the overall impact of different data augmentation methods on performance is relatively stable.

**Ablation study and Hyperparameter Study.** In the main text, due to space constraints, we have chosen to present representative experimental results from some selected datasets. In order to provide a more comprehensive demonstration of the effectiveness of SMART, we have included complete experimental results for eight datasets, including ablation experiments and hyperparameter experiments, in the appendix.

The correspondence between figures and experiment settings is as follows:

- Figure 13: Ablation Study of SMART on all datasets.
- Figure 14: Hyperparameter Study on proportional weight ratio $\lambda$ of SMART on all datasets.
- Figure 15: Hyperparameter Study on RNN dimension of SMART on all datasets.

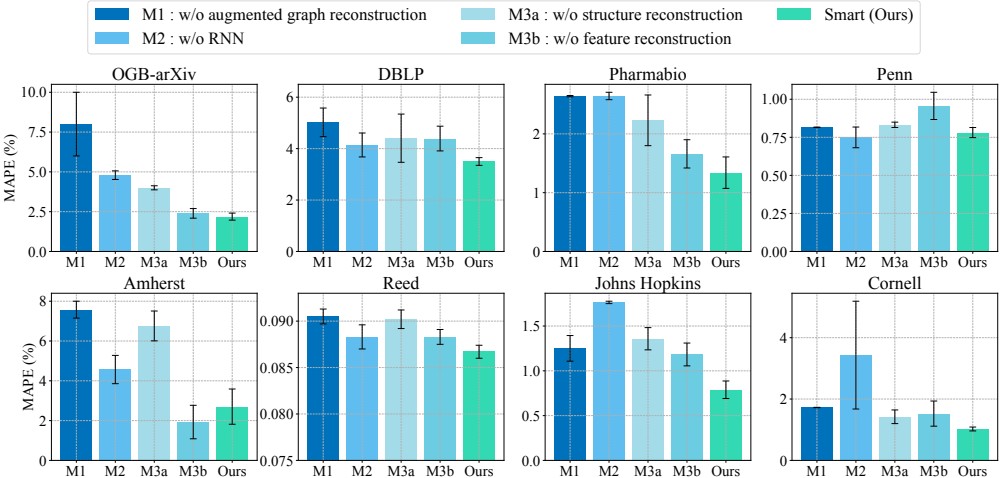

Figure 13: Ablation Study of SMART on all datasets.

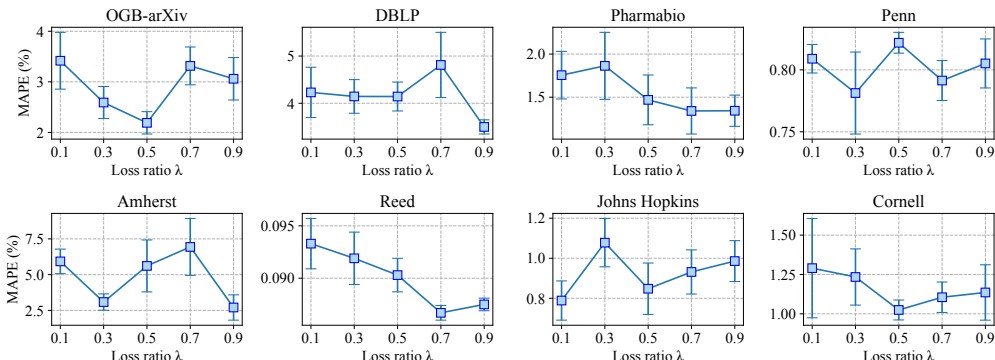

Figure 14: Hyperparameter Study on proportional weight ratio $\lambda$ of SMART on all datasets.

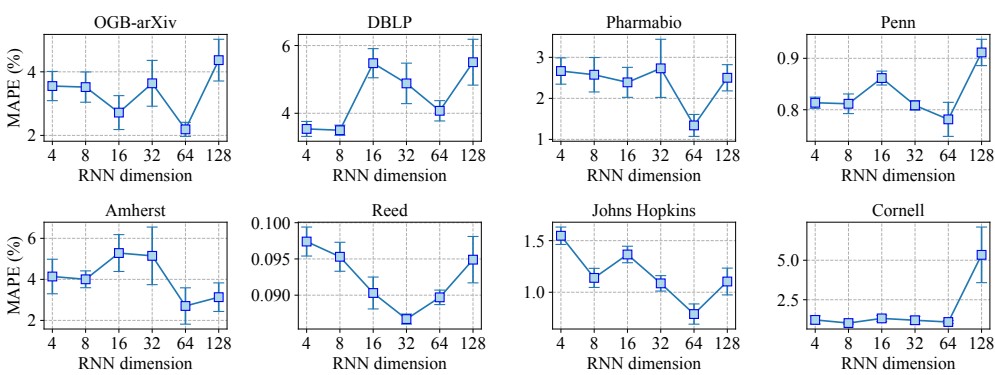

Figure 15: Hyperparameter Study on RNN dimension of SMART on all datasets.

