# OpenReview forum: "Temporal Generalization Estimation in Evolving Graphs"
_ICLR.cc/2024/Conference — ICLR 2024 poster_

### Official Review · Reviewer_e5fg · 2023-10-31

**Soundness:** 2 fair
**Presentation:** 3 good
**Contribution:** 2 fair
**Rating:** 5
**Confidence:** 4

**Summary:**

The paper analyzes and tackles the challenge of temporal representation distortion in GNNs as the graph evolves over time, so as to yield better generalization estimation. Based on a few assumptions, the authors theoretically establish that such distortion is strictly increasing over time with a lower bound. Furthermore, based on the pre-trained GNN and an RNN that performs temporal generalization estimation, the authors propose a solution that incorporates an adaptive feature extractor that operates through self-supervised graph reconstruction, aiming to estimate and adjust for the distortion. Under a synthetic random graph setting,  a closed-form generalization error is also derived. Experiments on synthetic and real-world benchmarks demonstrate the effectiveness of the proposed method in terms of substantially improving estimation accuracy and generalization ability.

**Strengths:**

1.	The general motivation for analyzing the generalization of evolving graphs is interesting and important. It is also interesting to see the theoretical analysis with a synthetic random graph demonstration with a closed-form generalization error.
2.	The presentation quality of this paper is in general satisfactory, in terms of the organization and figure demonstration.
3.	The experiments and analysis on the proposed method itself are relatively sufficient including model component ablation and hyper-parameter study.

**Weaknesses:**

1.	One major concern is the lack of baseline methods. The proposed method is only compared with a linear regression, which turns out to be a naive baseline. The methods mentioned in the related work D.2 are not compared. Moreover, only the MAPE with standard error is evaluated. More evaluation metrics are expected.
2.	It will be better if the methods are evaluated on a few more state-of-the-art backbones to further enhance the effectiveness of the proposed method.
3.	The assumptions made to conduct theoretical analysis and bound derivation could be a concern as they are often too strong. Evolving graphs also contain nodes disappearing in many cases(e.g. sensor networks with possible malfunction over time), which means the assumption of an ever-growing network can be often invalid. Moreover, the zero-mean requires a non-min-max preprocessing of features. The authors need to be specific about the claims.
4.	It would be better to sketch the connection between the information loss and the form in Equation 2, section 3.2.
5.	It would be better to define some notations such as the function l, and generalization error beforehand for better understanding of the idea.

**Questions:**

Please see the weaknesses.

**Details Of Ethics Concerns:**

N.A.

---

> ### Author Response · Authors · 2023-11-19
> **Response tot Reviewer e5fg**
>
> Thanks for your time and valuable comments that may help us for further improvement. We hope the following responses will serve to address your concerns.
>
> **Q1: Additional baselines and evaluation metrics.**
>
> R1: Thanks for your valuable suggestions. We have conducted additional two baselines (DoC and Supervised) with 4 classical evaluation metrics for regression task (MAPE, RMSE, MAE and Standard Error). The performance comparison is shown in Table 1 of the general response. Our SMART consistently outperforms both Linear and DoC, and closely approach the supervised methods. We provide more detailed information in Appendix H.3 in our revised paper. Regarding the methods discussed in related work in Appendix D.2, these methods focus on how to improve model performance on evolving graphs, instead of on how to characterize the model performance dynamics in graph evolution. Since the target is different,  these methods therefore can not be used for estimating generalization dynamics.
>
> **Q2: Performance on latest GNN backbone.**
>
> R2: Thanks for your comment. To fully validate and compare the effectiveness of different methods, we add Graph Transformer backbone besides GCN, GAT and GraphSAGE. As shown in Table 5, first of all, our SMART shows consistent best performance among different architectures. For different graph neural network architectures, they tend to capture the structure and feature information from different aspects. Due to the simple model structure of GCN, our SMART shows advanced prediction performance among other architectures, which is also consistent with the theory of classical statistical machine learning.
>
> Table 5. Performance comparison of different GNN architectures
>
> | Dataset | Model | GCN | GAT | GraphSAGE | TransformerConv |
> | --- | --- | --- | --- | --- | --- |
> | OGB-arXiv | Linear | 10.5224 | 12.3652 | 19.5480 | 14.9552 |
> |  | DoC | 13.5277±1.4857 | 12.2138±1.222 | 20.5891±0.4553 | 10.0999±0.1972 |
> |  | SMART | 2.1897±0.2211 | 3.1481±0.4079 | 5.2733±2.2635 | 3.5275±1.1462 |
> | DBLP | Linear | 16.4991 | 17.6388 | 23.7363 | 18.2188 |
> |  | DoC | 4.3910±0.1325 | 13.8735±4.1744 | 11.9003±1.8249 | 9.0127±2.6619 |
> |  | SMART | 3.4992±0.1502 | 6.6459±1.3401 | 9.9651±1.4699 | 6.4212±1.9358 |
> | Pharmabio | Linear | 32.3653 | 29.0404 | 31.7033 | 31.8249 |
> |  | DoC | 8.1753±0.0745 | 7.4942±0.0702 | 6.6376±0.0194 | 5.3498±0.2636 |
> |  | SMART | 1.3405±0.2674 | 1.2197±0.2241 | 3.1448±0.6875 | 2.7357±1.1357 |
>
> **Q3: Assumptions in theoretical analysis, including ever-growing and zero-mean normalized feature.**
>
> R3:  Thanks for your valuable suggestions. We generalize these two assumptions in the following way. (1) Regarding the ever-growing assumption of graphs, we modify it to the setting where nodes appearing after deployment have a probability of disappearing (Assumption 3-3). (2) Regarding the zero-mean normalized feature, we modify it to the setting where the feature vector has a constant mean conditioned on all previous graph states (Assumption 3-4). The completed assumptions please refer to Assumption 3 in Appendix C.2. Based on this assumption, we further extend the unavoidable representation distortion theorem to Theorem 4, and the proof of Theorem 4 can be found in Appendix C.4 and C.5.
>
> **Q4: The connection between information loss and graph reconstruction.**
>
> Thanks for your comments. Please refer to the response to Q3 in general response.
>
> **Q5: The definition of function $I$ and the generalization error.**
>
> R5: Thanks for your advice. In our paper, $I$ denotes the mutual information. For two random variable $X$ and $Y$, the mutual information $I$ is denoted as follows:
>
> $$I(X ; Y)=\sum_{x \in X} \sum_{y \in Y} p(x, y) \log \frac{p(x, y)}{p(x) p(y)}$$
>
> Generalization error refers to the performance of a trained model when making predictions on new, unseen data compared to the performance on the data it was trained on. In our work, the generalization error we estimate is the performance of the trained GNN model on the test samples during the graph evolution process. We have added the above definition to the revised paper, and we believe that this will be more helpful for readers to understand our work.

---

> > ### Author Response · Authors · 2023-11-22
> >
> > Thanks for your kind and helpful comments and we are looking forward to discussing with you to further improve our paper!

---

### Official Review · Reviewer_JYus · 2023-11-01

**Soundness:** 3 good
**Presentation:** 3 good
**Contribution:** 4 excellent
**Rating:** 8
**Confidence:** 3

**Summary:**

The paper presents an important question: how to ensure that GNN gives a good prediction for graphs that are rapidly evolving in time. The authors first give a theoretical proof to show that under mild conditions, as graph evolves, graph representation distortion is not avoidable (the loss is lower bounded). Secondly, the authors propose information losses in two phases: (1) information loss induced by RNN and (2) information loss induced by representation distortion. Then, the authors have proposed SMART that contains contrastive graph construction. The augmented feature graphs is randomly adding or dropping edges, and then structure reconstruction loss and feature reconstruction loss are used on the contrasted examples. The authors have verified the effectiveness of SMART on the barabasi-albert random graph, and performed extensive experiments to showcase SMART's effectiveness with evolving graphs.

**Strengths:**

S1. The paper targets evolving graph prediction, which is a challenging problem in the research community. It is novel to consider the evolving graph using information loss perspective, and the contrastive solution to resolve such problem is very convincing.

S2. The paper is clearly written. The authors give a clear problem definition with theoretical justification. There are several architecture and losses considerations in the methodology, yet each component is clearly addressed. The author has also provided a theoretical proof on the barabasi-albert random graph to showcase the effectiveness of SMART.

S3. The empirical results presented in the paper show that the proposed method SMART outperforms linear regression model, using the same GCN/GAT/GraphSage backbone structures and evaluated on four different datasets, showcasing the quality and robustness of SMART.

S4. Given the ubiquity and increasing reliance on evolving graph in real-world applications, the capability to adapt to evolving graph is important. SMART can adapt to various graph learning architectures and has potential impact for studying the changing graphs in time.

S5.  The ablation study is well-written and consider various loss configurations and hyper-parameter configueratons.

**Weaknesses:**

W1. The paper does not have a related work section. There should be at least some descriptions of the previous work which showcases that GNNs performances can suffer from the representation distortion over time, and the performance degradation occurs.

W2. There appears to be an inconsistency of mentioning the information loss in Section 3.2 and introducing SMART with contrastive loss in Section 3.3. The information loss is introduced, but not used in experiment settings (MPAE and standard error are used instead) or the later contrastive loss calculation. There should at least some experiments and measurements that connect information loss to contrastive learning post deployment, and to show that SMART is able to reduce the information loss induced by representation distortion. A theoretical justification is also fine.

W3. The data augmentation in graph is too naive. The authors have only considered to randomly add or drop edges, which should only work for graphs without edge labels. While the random data augmentation technique works for barabasi-albert random graphs, it should not be effective for the cases where edge labels are also changing (for example, people change their relationship status with other people). The authors should consider more complex data augmentation techniques in graphs.

**Questions:**

Q1. Theorem 1 is evaluated only for one layer GNN with a Leaky ReLU activation. Is Theorem 1 adjustable to multi-layer GNN and different GNN backbone structures such as GCN/GAT/GraphSage?

Q2. Figure 3, what is the y-axis prediction loss? Why not use other evaluation metrics to showcase the GNN prediction performances deterioration?

Q3. Is there a particular reason that RNN is used instead of other time-series model to capture the temporal variation?

Q4. Will the contrastive graph reconstruction improve the feature extractor during the pre-deployment phase? Why just limit the contrastive learning and reconstruction to post-deployment?

Q5. How will the proposed algorithm be used towards more complex real-time graphs? For example, how to deal with graphs that contain changing node labels and changing edge labels? How to apply the methods to heterogeneous graphs or spatial temporal graphs?

---

> ### Author Response · Authors · 2023-11-19
> **Response to Reviewer JYus (1/2)**
>
> Thanks for appreciating our work and give very detailed reviews. We are greatly encouraged that you appreciated our contributions including novel and convincing methods, problem significance, clear writing and potential impact of our study. We hope the following response will serve to address your concern and improve your confidence.
>
> **Q1: Supplement to related work.**
>
> R1: Thanks for your comments. Please refer to the response to Q4 in general response.
>
> **Q2: Clarification of information loss by representation distortion and graph reconstruction.**
>
> R2: Thanks for your comments. Please refer to the response to Q3 in general response.
>
> **Q3: More data augmentation techniques in graphs.**
>
> R3: We have added three graph data augmentation methods to generate different views for self-supervised graph reconstruction in a contrastive manner, i.e., DropEdge [5], DropNode [6] and Feature Mask [7]. Table 3 shows the experimental results, and our SMART shows the best performance on three datasets. We observe that different data augmentation methods on performance have similar performance.
>
> Table 3. Performance comparison with different graph data augmentation methods.
>
> | Method | OGB-arXiv | DBLP | Pharmabio |
> | --- | --- | --- | --- |
> | Linear | 10.5224 | 16.4991 | 32.3653 |
> | DoC | 9.5277±1.4857 | 4.3910±0.1325 | 8.1753±0.0745 |
> | SMART (DropEdge) | 2.1897±0.2211 | 3.4992±0.1502 | 1.3405±0.2674 |
> | SMART (DropNode) | 1.7163±0.3783 | 3.8166±0.5124 | 0.9050±0.2606 |
> | SMART (Feature Mask) | 1.8733±0.2729 | 3.0183±0.4073 | 0.7352±0.4001 |
>
> **Q4: Extension of theoretical analysis for more complex GNN model.**
>
> R4: Thanks for your comments. As we mentioned in the general response, we generalize Theorem 1 to a more general class of multi-layer graph neural network architectures (i.e., Theorem 3). Specifically, we consider the model where a feedforward neural network of an arbitrary depth is concatenated with the graph convolution network. The detailed theoretical analysis please refer to Appendix C.1 and the proof can be found in Appendix C.3.
>
> **Q5: Explanation of Figure 3.**
>
> R5: Thanks for your comment. Y-axis indicates the prediction loss of GNN model on test datasets during graph evolution. It is also possible to use other performance indicators, while the loss on the test sample is just a simple and intuitive indicator.
>
> **Q6: The selection of time series models to capture the temporal variations.**
>
> R6: To capture the temporal dynamics of GNN generalization variation, we propose an RNN-based method (i.e. LSTM) and achieve satisfied estimation performance. In our paper, we used the vanilla LSTM model to demonstrate the importance of temporal encoding via time series models. We do agree that under a more complex scenario, advanced time series model may achieve better performance.
>
> To further investigate this problem, we additionally replace the LSTM with other two time series model Bi-LSTM and TCN. As depicted in Table 4, our SMART achieves the best performance among three different time series models on both OGB-arXiv and Pharmabio datasets. On DBLP dataset, SMART achieves the best performance using LSTM, while SMART equipped with Bi-LSTM and TCN show comparable performance with DoC. In order to maintain consistency in the experiment, we adopted the same hyperparameter settings. From the perspective of parameter size, the parameter quantity of Bi-LSTM is nearly twice that of LSTM, while the parameter quantity of TCN is very small. We observe that as the parameter size increases, the prediction error generally decreases first and then increases. From these preliminary results, it is indeed important to choose appropriate models and parameter settings based on the complexity of the data.
>
> Table 4. Performance comparison with different time series model.
>
> | Method | OGB-arXiv | DBLP | Pharmabio |
> | --- | --- | --- | --- |
> | Linear | 10.5224 | 16.4991 | 32.3653 |
> | DoC | 9.5277±1.4857 | 4.3910±0.1325 | 8.1753±0.0745 |
> | SMART (LSTM) | 2.1897±0.2211 | 3.4992±0.1502 | 1.3405±0.2674 |
> | SMART (Bi-LSTM) | 4.3601±0.5816 | 4.2402±0.6996 | 1.1219±0.0397 |
> | SMART (TCN) | 1.5725±0.2500 | 4.5970±0.2607 | 1.8830±0.4161 |

---

> > ### Author Response · Authors · 2023-11-19
> > **Response to Reviewer JYus (2/2)**
> >
> > **Q7: Contrastive learning and graph reconstruction in pre-deployment phase.**
> >
> > R7: Thanks for your advice. Sorry for the vague explanation we may have in the previous version. In SMART, we also conduct contrastive learning and graph reconstruction during training in pre-deployment phase, as shown in Algorithm 1 of Appendix A. The self-supervised loss function plays an important role to optimize a better estimator. We have further clarified the use of contrastive learning and graph reconstruction in our revised paper.
> >
> > **Q8: Extension to more complex graphs, e.g., graphs where labels may change, heterogeneous graphs, spatial temporal graphs.**
> >
> > R8: Thanks for your valuable comments. These are all promising directions for future work. But we also would like to point out that the setting considered in this paper, where only the graph topology and node feature evolve, is one of the most basic settings in analyzing graph evolution. In our paper, we establish both  theoretical and experimental results showing that even without change of labels and in graphs without requiring heterogeneity, representation distortion is inevitable. We do believe that our proposed SMART can be fused with more powerful contrastive learning or time series techniques and  adapted  to more complicated graph evolution scenarios.
> >
> > [5] Yu Rong, et al., "DropEdge: Towards Deep Graph Convolutional Networks on Node Classification", ICLR, 2020.
> >
> > [6] Pál András Papp, et al. "DropGNN: Random Dropouts Increase the Expressiveness of Graph Neural Networks", NeurIPS, 2021.
> >
> > [7] Weihua Hu, et al. "Strategies for Pre-training Graph Neural Networks", ICLR 2020.

---

> > > ### Comment · Reviewer_JYus · 2023-11-21
> > > **Response to the authors**
> > >
> > > Thank you for the explanations. My concerns are mostly addressed and I would recommend this paper for acceptance.

---

> > > > ### Author Response · Authors · 2023-11-22
> > > > **Response to Reviewer JYus**
> > > >
> > > > Thanks for your kind comment and helpful suggestions. We appreciate for your recommendation for acceptance. We look forward to further discussion with all reviewers.

---

### Official Review · Reviewer_qkRe · 2023-11-02

**Soundness:** 2 fair
**Presentation:** 3 good
**Contribution:** 2 fair
**Rating:** 6
**Confidence:** 3

**Summary:**

This paper proposes a method (SMART) for performing generalization error estimation on temporal node classification tasks. The authors show that representations will become increasingly distorted as time increases, and propose to use both a structure graph reconstruction and feature construction loss to improve representation quality. The adapted features are feed into an RNN that is trained to predict the loss. The authors also propose two theorems: one which shows that distortion is strictly increasing over time, and the other one is specific to the Barbasi Albert graph and is used to argue the benefits of SMART.

**Strengths:**

- This paper studies an interesting problem (generalization gap prediction for temporal node classification) that has not been well-studied in the graph representation learning literature before.

- The authors conduct experiments across a variety of datasets, and conduct several ablations to demonstrate the benefits of each of the components of SMART.

- The authors attempt to provide some theoretical bounds and use these, as well as an information theory framework, to ground the proposed method.

**Weaknesses:**

- The authors only compared to a single linear regression baseline. While other methods for generalization prediction are not specific to temporal graph data, it should still be possible to consider some. Maybe some from [1] could be adapted? In this vein, I think there should be more citations to other generalization error predictors and a dedicated related works section in the main paper.

- The theorems need to be discussed more. For example, in theorem 1, I don't think that beta has been defined. Furthermore, while the authors make the strong but acceptable assumption that the graph will only add edges, I was wondering if it was also assumed that the underlying graphs were homophilous. Do the authors expect the proposed method to work on heterophilus graphs as well?

- The writing could be improved. For example, its not clear why the approach is referred to as "constrastive." From my understanding the authors do augment the graph, but the overall loss is purely reconstruction based.

[1] Predicting with Confidence, Guillory et al. ICCV 2021(https://arxiv.org/abs/2107.03315)

**Questions:**

Please see the weaknesses above.

I'd appreciate some clarifications about the theorems and potential baselines as mentioned above.

---

> ### Author Response · Authors · 2023-11-19
> **Response to Reviewer qkRe**
>
> Thanks for your time and constructive suggestions. We are pleased for your positive comments. In the response below, we provide answers to your questions in order to address the concerns and increase your confidence.
>
> **Q1: Model comparison with other baselines.**
>
> R1: Thank you very much for pointing out a relevant paper [1] in the field of computer vision. After in-depth reading, we adopt the DoC method proposed in this paper as a baseline for performance comparison. We present some additional experimental results in Q2 of the general response, where 4 evaluation metrics are shown in Table 1. We observe that in our experimental settings,  SMART shows consistent better performance than DoC. This may be due to the reason that DoC method is optimized for the setting where the testing distribution slightly differs from training data. However, in our setting, graph evolution is very fast and the evolving graphs are usually very different from their initial states. We also provide more detailed discussion in Appendix H.3 in our revised paper.
>
> [1] Devin Guillory, et al. "Predicting with Confidence on Unseen Distribution." ICCV, 2021.
>
> **Q2: The definition of beta and the consideration of homogeneity assumption of graphs in theoretical analysis.**
>
> R2: Thanks for your valuable advice. In our theoretical analysis, $\beta$ is the slope ratio for negative values instead of a flat slope in ReLU. We have added an explanation of $\beta$ in our revised paper. Regarding the homogeneity of graphs, we do not make any assumptions in both theoretical analysis and empirical study. Meanwhile, we conduct a data analysis of the changes of homogeneity ratio over time on all the datasets as shown in Figure 10 of Appendix G. We observe that different datasets exhibit diverse variations in the homogeneity ratio of graphs and that SMART consistently outperforms the other two self-supervised baselines in all cases.
>
> **Q3: Explanation of contrastive way in self-supervised graph reconstruction.**
>
> R3: Constrastive self-supervised learning is a popular and effective learning strategy to enrich the supervised signals. It can be divided into two stages: first is the graph augmentation and then is the contrastive pretext tasks. Therefore, a general learning objective of contrastive self-supervised learning approaches is denoted as follows:
>
> $$\theta^*, \varphi^* = \arg \min_{\theta, \varphi} \mathcal{L}(p_{\varphi}(f_{\theta}(\tilde{A}^{(1)}, \tilde{X}^{(1)}), f_{\theta}(\tilde{A}^{(2)}, \tilde{X}^{(2)})))$$
>
> where $\tilde{A}^{(1)}$ and $\tilde{A}^{(2)}$ are two augmented graph adjacency matrices, $\tilde{X}^{(1)}$ and $\tilde{X}^{(1)}$ are two node feature matrices under different augmentations. In our setting, We perform data augmentation (e.g., randomly dropout node, dropout edge, mask feature) on the graph data with a probability of 0.5, keeping it unchanged with a probability of 0.5. Afterwards, we define a pretext task is to minimize the distance of two augmented graphs via link prediction and node feature reconstruction. We note that our proposed SMART is actually a special setting in the general contrastive learning framework.

---

> > ### Author Response · Authors · 2023-11-22
> >
> > Thanks for your kind and helpful comments and we are looking forward to discussing with you to further improve our paper!

---

> > > ### Comment · Reviewer_qkRe · 2023-12-04
> > > **Acknowledgement of Response**
> > >
> > > Hello Authors.
> > >
> > > I have read the rebuttals, and your reviews. I appreciate your efforts and adjusted my score as a decent portion of my comments were addressed. Thanks for adding the DoC baseline. However, I strongly encourage that other baselines for predicting generalization are added and cited. For example, there are invariance based methods and simple ensemble/model agreement that would be able to serve as naive baselines. While I suspect that SMART should beat them, I think its importance to include these as this is one of the early works in the graph setting.
> > >
> > > Also, I still think that the writing should still be improved. I'm well aware of what contrastive learning is; I found the *presentation* of the loss unclear.

---

### Official Review · Reviewer_btk3 · 2023-11-03

**Soundness:** 3 good
**Presentation:** 3 good
**Contribution:** 3 good
**Rating:** 5
**Confidence:** 5

**Summary:**

This paper studies the problem of estimating the generalization performance of GNNs on evolving graphs. The authors propose SMART, which estimates the generalization performance of GNNs without the need for manual annotation post-deployment. SMART employs self-supervised contrastive graph reconstruction to update the feature extractor, minimizing information loss during the dynamic evolution process.

**Strengths:**

S1. The problem is of significant importance in dynamic graph learning, and the authors provide theoretical proofs to demonstrate that representation distortion is inevitable.

S2. The authors derive a closed-form expression of the generalization error bound on synthetic data and verify the effectiveness of the proposed method.

S3. The paper is well-written, and I have no complaint about the presentation of the paper.

**Weaknesses:**

W1. The theoretical analysis is limited to single-layer GCN models. It would be nice to see an extension to more complex models.

W2. The baseline only includes simple linear regression models. However, a better baseline should be a model that continuously acquires new node labels and gets retrained on the new data, to examine whether the generalization curve of such retraining is close to the prediction of the proposed method. This can more intuitively demonstrate the advantages of the method.

W3. The paper shows that in the citation networks, the performance drop of pretrained models is mainly due to the increase of nodes and edges, while the category distribution does not change significantly. However, in real-world applications, changes in labels are very common, which will lead to concept drift and cause the model's performance to decrease dramatically. It is worth discussing whether the proposed method is applicable to such scenarios with frequent label changes and concept drift.

**Questions:**

See W1-W3 for details.

---

> ### Author Response · Authors · 2023-11-19
> **Response to Reviewer btk3**
>
> Thanks for your time and valuable comments. We are encouraged that you appreciated our technical contributions including problem significance, theoretical proof and well-written presentation. In the response below, we provide answers to your questions in order to address the concerns.
>
> **Q1: Extension of theoretical analysis to a more complex scenario.**
>
> R1: Thanks for your comments. As we mentioned in the general response, we generalize Theorem 1 to a more general class of multi-layer graph neural  network architectures (i.e., Theorem 3). Specifically, we consider the model where a feedforward neural network of an arbitrary depth is concatenated with the graph convolution network. For the detailed theoretical analysis, please refer to Appendix C.1 and the proof can be found in Appendix C.3. We also generalize the assumption that allows missing nodes and non-zero feature bias in  evolution.
>
> **Q2: More model comparisons, e.g. a supervised manner.**
>
> R2: Thanks for your suggestion. We have added a supervised baseline and compared it with our SMART. Please refer to Q2 in general response. We also provide more detailed information in Appendix H.3 in our revised paper. Overall, SMART achieves comparable performance on dataset OGB-arXiv, while achieving slightly worse performance on dataset Pharmabio, when compared to the supervised baseline.
>
> **Q3: Generalizability of SMART for label changes and concept drift.**
>
> R3: We agree with the reviewer that the graph topology $A$, feature vectors $X$ on all nodes and even labels $Y$ can change during the graph evolution. In other ways, the joint distribution $P(A,X,Y)$ can change during the graph process. Using the definition of the conditional probability, we must have   $P(A,X,Y)=P(Y|A,X)P(A,X)$. This implies that the change of joint distribution $P(A,X,Y)$ can be due to the change of $P(Y|A,X)$,$P(A,X)$or  both of them. In this paper, we aim at the problem where only $P(A,X)$changes and theoretically showed the inevitable representation distortion even without of change of $P(Y|A,X)$. We proposed the algorithm SMART mainly to address the case where only $P(A,X)$ and thus do not consider the setting where the label changes and concepts drift. But we do believe techniques focusing on the changing labels can be fused with our algorithm for handling the case where both distribution $P(Y|A,X)$ and $P(A,X)$ evolve.

---

> > ### Author Response · Authors · 2023-11-22
> >
> > Thanks for your kind and helpful comments and we are looking forward to discussing with you to further improve our paper!

---

### Official Review · Reviewer_jJDp · 2023-11-04

**Soundness:** 3 good
**Presentation:** 3 good
**Contribution:** 3 good
**Rating:** 6
**Confidence:** 3

**Summary:**

This study investigates the representation distortion of graphs during evolution. The authors proposed SMART to estimate the temporal generalization performance of GNN. The authors provided theoretical proofs as well as numerical experiments, showing the distortion of representation is inevitable and providing a way of estimating generalization loss.

**Strengths:**

This is a novel study trying to analyze the representation distortion from an information theory perspective, which I think is an important and interesting question.  The authors theoretically establish a lower bound and conducted various numerical experiments using both synthetic datasets and real world datasets. Introduction is straightforward. The results seem promising. The ablation study is convincing and sufficient.

**Weaknesses:**

The authors mainly compared their results with linear regression. The authors may add more model comparisons to make the results more convincing.

**Questions:**

Figure 1 is a bit confusing: what’s the x axis? And what are the nodes at each year?

For the information loss, the authors considered the loss from RNN as well as from representation distortion. Can the authors comment on the loss of information from graphs to their low dimensional representations and their effects on the model.

More evaluation metrics besides MAPE should be considered for the synthetic datasets.

Interestingly GCN seems achieved the best performance in the three compared GNNs, can the authors comment on the contribution of different GNN structures to the performance?

---

> ### Author Response · Authors · 2023-11-19
> **Response to Reviewer jJDp**
>
> Thank you for your positive and constructive comments. We are pleased that you acknowledged the novelty and significance of our focused problem, promising results and straightforward presentation. In the following response, we answer your comments/questions point by point.
>
> **Q1: More model comparisons to make the results more convincing.**
>
> R1: We have added two additional baselines, DoC and Supervised, and conducted performance comparison on 3 datasets. The experimental results are shown in the general response (Table 1), and we also provide more detailed experimental results (Table 9-10, Figure 12-16)  in Appendix H.3 in our revised paper.
>
> **Q2: Clarify the meaning of Figure 1.**
>
> R2: Thanks for your valuable suggestion. Figure 1 depicts the changes of the graph scale and GCN classification performance nearly 30 years on Pharambio dataset. Therefore, the x axis is the range of year. Each node in the graph means a paper in an academic co-authorship graph.
>
> **Q3: The impact of information loss in graph representation on the generalization estimator.**
>
> Thanks for your comments. Please refer to the response to Q3 in general response.
>
> **Q4: More evaluation metrics for the synthetic datasets.**
>
> R4: We have added MAPE, RMSE and MAE on the synthetic datasets. The results are presented in following Table 2. The detailed analysis please refer to Appendix F. Since the DoC method only applies to classification problems using the average confidence, we thus do not compare it with our SMART on the synthetic setting.
>
> Table 2. Performance comparison on different Barabási–Albert graph setting.
> | Barabási–Albert (BA) Random Graph |  | Linear |  |  | SMART (Ours) |  |  |
> | --- | --- | --- | --- | --- | --- | --- | --- |
> |  |  | MAPE | RMSE | MAE | MAPE | RMSE | MAE |
> | BA ($\mathcal{N}_0$ = 1000) | $m=2$ | 79.2431 | 0.0792 | 0.0705 | 7.1817±1.2350 | 0.0057±0.0008 | 0.0042±0.0006 |
> |  | $m=5$ | 74.1083 | 0.0407 | 0.0398 | 4.2602±0.5316 | 0.0039±0.0004 | 0.0035±0.0004 |
> |  | $m=10$ | 82.1677 | 0.1045 | 0.0925 | 9.1173±0.1331 | 0.0077±0.0010 | 0.0071±0.0009 |
> | Dual BA ($\mathcal{N}_0$ = 1000, $m_1$=1) | $m=2$ | 61.8048 | 0.0676 | 0.0615 | 7.9038±1.8008 | 0.0088±0.0022 | 0.0069±0.0017 |
> |  | $m=5$ | 67.6442 | 0.0827 | 0.0771 | 3.8288±0.1706 | 0.0049±0.0013 | 0.0040±0.0010 |
> |  | $m=10$ | 38.4884 | 0.0298 | 0.0297 | 1.9947±0.1682 | 0.0026±0.0002 | 0.0023±0.0003 |

---

> > ### Author Response · Authors · 2023-11-22
> >
> > Thanks for your kind and helpful comments and we are looking forward to discussing with you to further improve our paper!

---

### Author Response · Authors · 2023-11-19
**General Response by Authors (1/2)**

Dear area chair and reviewers,

We appreciate the reviewer's time and valuable comments. Overall, the reviewers deem that we investigate an important, interesting (all 5 reviewers) and not well-studied (qkRe, JYus) question, recognize our work is a novel study (jJDp, JYus) with promising results (jJDp) and potential impact (JYus) for future work. Moreover, the experiment study is convincing, sufficient (jJDp, JYus, e5fg) and whole paper is well-written (btk3, JYus).

However, we notice that some reviewers may have potential concerns about additional experimental comparison, assumptions in theoretical analysis, etc. In the general response, we first present an overview clarification of some common suggestions. In the following individual response, we provide detailed answers to all the specific comments/questions raised by the reviewers, and supplement new experimental results for further demonstration of the effectiveness of our methods. We hope our responses will serve to address reviewers' concerns.

**Q1: Extension of Theoretical Analysis.**

R1: We generalize the Theorem 1 into more general scenarios: (1) more general class of multi-layer graph neural  network architectures, (2) graph evolution process with disappearing nodes and (3) non-zero bias of node features. We first consider the model where a FNN of an arbitrary depth is concatenated with a GCN. We show that the expected deviation $\ell_{\tau}(i)$ is lower bounded by a function that is strictly increasing with respect to the time $\tau$. A detailed theoretical analysis can be found in Theorem 3 in Appendix C, while the proof can be found in Appendix C.3. Next, we allow missing nodes and evolution with bias in our assumption and show a similar lower bound. Please refer to Assumption 3 and Theorem 4 in Appendix C.2.

**Q2. Additional model comparison and experimental analysis.**

R2. We have added two additional baselines (DoC [1] and Supervised-manner as suggested by Reviewer btk3) based on the suggestions of the reviewers under 4 metrics (MAPE, RMSE, MAE with Standard Error) as shown in Table 1. Details of baselines and metrics are presented in Appendix H.1. Additional results are presented in Appendix H.3.

(1) We observed that SMART consistently outperforms the other two self-supervised methods (linear regression and DoC) on different evaluation metrics, demonstrating the superior temporal generalization estimation of our methods.

(2) We compared SMART with a supervised baseline. This method requires new node labels and retrain the model on the new data after deployment, which is very different from our self-supervised method SMART that requires no new human annotated labels after deployment. On OGB-arXiv dataset, our SMART achieves comparable performance with Supervised, while on a harder dataset Pharmabio, supervised baseline consistently outperforms than our SMART.

(3) We added additional experiments on different time series models (LSTM, Bi-LSTM and TCN), GNN backbones (GCN, GAT, GraphSAGE and TransformerConv) and graph augmentation methods (DropEdge, DropNode and Feature Mask) on SMART. The details of additional experiments please refer to Appendix H in our revised paper.


Table 1. Performance comparison of SMART and baselines on three academic network datasets.

| Dataset | Metric | Linear | DoC | SMART (Ours) | Supervised |
| --- | --- | --- | --- | --- | --- |
| OGB-arXiv | MAPE | 10.5224 | 9.5277±1.4857 | 2.1897±0.2211 | 2.1354±0.4501 |
|  | RMSE | 0.4764 | 0.3689±0.0400 | 0.1129±0.0157 | 0.0768±0.0155 |
|  | MAE | 0.4014 | 0.3839±0.0404 | 0.0383±0.0083 | 0.0218±0.0199 |
| DBLP | MAPE | 16.4991 | 4.3910±0.1325 | 3.4992±0.1502 | 2.5359±0.4282 |
|  | RMSE | 0.5531 | 0.1334±0.0058 | 0.1165±0.0444 | 0.0914±0.0175 |
|  | MAE | 0.4310 | 0.1162±0.0310 | 0.0978±0.0344 | 0.0852±0.0038 |
| Pharmabio | MAPE | 32.3653 | 8.1753±0.0745 | 1.3405±0.2674 | 0.4827±0.0798 |
|  | RMSE | 0.7152 | 0.1521±0.0014 | 0.0338±0.0136 | 0.0101±0.0015 |
|  | MAE | 0.6025 | 0.1521±0.0013 | 0.0282±0.0120 | 0.0088±0.0026 |

---

> ### Author Response · Authors · 2023-11-19
> **General Response by Authors (2/2)**
>
> **Q3. Further clarification of information loss by representation distortion and graph reconstruction.**
>
> We analyzed the representation distortion via the information loss and showed that the zero information loss implies zero reconstruction error. To see this, by data processing inequality, we have $I(X;Y)\ge I(\varphi(X);Y)$ for any function $\varphi$. By the property of mutual information, equality holds if and only if the function $\varphi$ is a bi-jective function. This further indicates that
>
> $$\min_{\phi}\mathbb{E}[\|X-\phi\circ\varphi(X)\|^2]=0.$$
> In our paper, the information loss induced by the representation distortion is
>
> $$I({G(A_\tau,X_\tau)\}^k_{\tau = t_{deploy}+1},\mathcal{D};\ell_k) - I(\{\varphi \circ G(A_\tau,X_\tau)\}^k_{\tau=t_{deploy}+1},\mathcal{D};\ell_k) \ge 0.$$
>
> To minimize the information loss, i.e., achieve equality, we have to choose an appropriate $\varphi$ such that $\varphi$ is a bijective map. By information theory, equality further indicates that
>
> $$\min_{\phi}\sum_{\tau}\mathbb{E}\|G(\tau, X_\tau)-\phi\circ\varphi\circ G(A_\tau, X_\tau) \|_2^2=0.$$
>
> This can be viewed as the setting where the minimum reconstruction error is zero. We have revised the paper to clarify the relationship between zero information loss and optimal reconstruction.
>
> **Q4. Supplement to related work.**
>
> R4. In Appendix E, we have discussed the related work about distribution shift estimation and evolving graph representation. In order to provide a clearer overview of the relevant work, we have added a brief summary in the introduction section of our revised paper.
>
> **Summary of the Paper Revisions**
>
> The main updates are summarized as follows:
> 1. Page 1, Sec. 1, add an overview of related works about distribution shift evalution.
> 2. Page 4, Sec. 2.3, add an extension of theoretical analysis about GNN representation distortion.
> 3. Page 5, Sec. 3.2, highlight the relationship between information loss and graph reconstruction.
> 4. Page 6, Sec 3.3, highlight the usage of graph reconstruction in pre-deployment phase.
> 5. Page 8, Sec 5.2, add more baseline comparison and experimental analysis.
> 6. Page 16-22, we add a new section Appendix C to present the detailed theorem and theoretical proof for extensions on Theorem 1.
> 7. Page 26, Appendix F, we add the complete experiment results on synthetic BA random graph datasets.
> 8. Page 27-32, we re-write Appendix H, which provides a detailed explanation of our experimental details and analysis, including baseline experiments, comparison of different GNN backbones, time series models, graph data augmentation methods, etc.

---

### Meta-Review · Area_Chair_Vq12 · 2023-12-04

**Metareview:**

This paper establishes a theoretical lower bound on representation distortion of Graph Neural Networks (GNNs) over time. The paper proposes a self-supervised graph reconstruction method, which outperforms conventional approaches in estimating temporal distortion on synthetic and real-world evolving graphs.

In general, the reviewers find that this is a novel study that tries to answer interesting questions. Several reviewers found initially that more model comparisons would strengthen the paper (the baseline only included simple linear regression models). In response, the authors added some additional baselines.

As pointed out by some reviewers, it would be beneficial to add more discussion about the results of the theorems. Various more minor comments about the presentation were made.

Overall, I think the contribution of the paper is good. The problems raised by the reviewers are of a more minor nature and can be improved in the camera-ready version. I therefore recommend acceptance.

**Justification For Why Not Higher Score:**

This is a borderline case, but I think it makes some interesting contributions from a theory point of view (although I have to admit that this is not my area of expertise).

**Justification For Why Not Lower Score:**

This is a borderline case, but I think it makes some interesting contributions from a theory point of view (although I have to admit that this is not my area of expertise).

---

### Decision · Program_Chairs · 2024-01-16

Accept (poster)